

**New derivation and interpretation of the complementary relationship for**
**evapotranspiration**
Sha Zhou[1,2*], Bofu Yu[3]
[1]State Key Laboratory of Earth Surface Processes and Hazards Risk Governance (ESPHR),
Faculty of Geographical Science, Beijing Normal University, Beijing, China
[2]Institute of Land Surface System and Sustainable Development, Faculty of Geographical Science,
Beijing Normal University, Beijing, China
[3]School of Engineering and Built Environment, Griffith University, Nathan, Queensland, Australia
*Correspondence author. Email: shazhou21@bnu.edu.cn
**Abstract.** The complementary relationship (CR) between actual evapotranspiration (ET) and
apparent potential evapotranspiration (PETa) is widely used as a simple yet effective method for
ET estimation. However, most existing CR formulations are empirical, lacking rigorous derivation
based on physics. In this study, the complementary relationship was derived analytically with a
physically meaningful parameter: the wet Bowen ratio, defined as the Bowen ratio when the
surface becomes saturated. This parameter can be computed from observations without calibration.
Fundamentally, the CR is shown to originate from partitioning of the net radiation, with ET directly
linked to the latent heat and PETa proportional to the sensible heat. Additionally, ET is linearly
related to and constrained by the energy-based potential evapotranspiration (PETe). The
physically-based relationship among ET, PETa, and PETe has important implications for our
understanding of the spatial and temporal variations in ET and would promote practical application
of the complementary relationship for ET estimation across different environments.



## 1. Introduction

**1. Introduction**
Terrestrial evapotranspiration (ET) plays a vital role in land-atmosphere exchanges of water,
energy, and carbon fluxes and thereby influences weather and climate as well as the water and
carbon cycling (Ault, 2020; Gentine et al., 2019; Miralles et al., 2019; Zhou et al., 2022, 2023).
While many approaches have been attempted to estimate ET, the complementary relationship (CR)
for ET, which was first proposed by Bouchet (1963) and operationalized by Morton (1983),
provides a simple conceptual framework for estimating ET using routine meteorological
observations (Han and Tian, 2020; Ma et al., 2021; Zhang and Brutsaert, 2021).

The CR essentially describes the relationship between three types of ET over land surface
(Brutsaert, 2015). The first type is the actual ET, which is the water vapor flux from open water,
soil, and vegetation over an area. The second type is the potential ET (PET), i.e., the ET that would
occur from the same area with the same net radiation, but with unlimited water supply, i.e., a
saturated evaporative surface. As the PET is limited by available energy, it is termed as PETe in
this study. The third type is the apparent potential ET (PETa), which is the ET that would occur
from a small, saturated surface within a large and unsaturated area, with abundant energy supply
from the net radiation and the surrounding environment (Zhou and Yu, 2024). It is assumed that
the saturated evaporative surface is too small to affect aerodynamic conditions, i.e., air temperature,
humidity, and wind speed. PETa therefore depends on the prevailing aerodynamic conditions over
the large dry area. In practice, PETe is represented by the actual ET over a large lake or reservoir,
while PETa by the actual ET measured with a small evaporation pan placed in an otherwise dry
environment (Kahler and Brutsaert, 2006). These three types of ET converge when the large area
is saturated everywhere. As the large area dries up, ET is limited by available water and falls below



PETe. Simultaneously, a lower ET reduces the moisture content of the atmosphere and reduced
evaporative cooling increases air temperature, resulting in a warmer and drier atmosphere with a
higher PETa than PETe. These processes lead to a complementary relationship between ET and
PETa when water supply is limited.

Based on the conceptual framework above, a series of CRs have been formulated, with most being
empirical, and none of them is fully physically-based (Bouchet, 1963; Brutsaert and Parlange,
1998; Brutsaert, 2015; Crago et al., 2016; Granger, 1989; Szilagyi, 2007; Szilagyi et al., 2017,
2022; Tu et al., 2023). This is because the physical mechanisms underlying the CR and the
quantitative relationship between the three types of ET remain unclear, which hinder our
understanding of the CR and the derivation of a physically-based CR for accurate estimation of
ET. The original CR proposed by Bouchet (1963) assumes that the relationship is symmetric with
respect to energy conservation, which is, however, not supported by observations, and several
alternative CRs have been proposed with an empirical parameter to account for the departure from
the original symmetric CR (Brutsaert and Parlange, 1998; Brutsaert, 2015; Crago et al., 2016;
Granger, 1989; Szilagyi, 2007). Due to a lack of physical interpretation of the parameter involved,
its values must be estimated through calibration with location-specific observations, which hinders
and limits the application of these alternative CRs for ET estimation.

In addition, a lack of definitive estimators of PETe and PETa also hinders development of a
physically-based CR (Crago et al., 2016; Tu et al., 2023). As PETe cannot be directly measured
unless water supply is unlimited all the time, previous studies used the Priestley-Taylor equation
(Priestley and Taylor, 1972) to approximate PETe, which is, however, quite different from



observed ET over the ocean, indicating that the Priestley-Taylor equation is not entirely appropriate
for estimating PETe (Yang and Roderick, 2019; Zhou and Yu, 2024). On the other hand, PETa is
generally measured using an evaporation pan or estimated from the Penman equation (Penman,
1948), which is, however, not consistent with the definition of PETa, as the Class A evaporation
pan is large enough to have measurable effect on the air temperature and humidity around the pan
and the Penman equation does not consider energy transferred from its surrounding environment
(Brutsaert, 2015; Kahler and Brutsaert, 2006). An in-depth understanding of the physical processes
underlying the complementary relationship and accurate estimation of the three types of ET are
therefore crucial for formulating a physically sound complementary relationship.

The objective of this paper is to reappraise the physical foundation of the complementary
relationship and derive a physically-based CR for estimating ET over land. Based on theoretical
reasoning and analysis, we identify the key physical processes underlying the complementary
relationship. By estimating the three types of ET based on their definitions and physical processes,
we formulate an alternative CR with a physically meaningful parameter. This study would advance
our understanding of the physics behind the complementary relationship to support its practical
application for ET estimation over land.

**2. Concept of the complementary relationship and empirical formulations**
When the land surface is well supplied with water, the magnitude of the three types of ET are
identical, depending on the energy supply and aerodynamic conditions.

91                          $$ET = PET_e = PET_a \qquad\qquad (1)$$



As the moisture supply at the evaporative surface decreases, the available water is insufficient to
meet the evaporative demand, i.e., $PET_e$, and the energy not expended on $ET$ is shifted to be
sensible heat ($H$) which increases with the difference between surface and air temperatures and
causes $PET_a$, e.g., the evaporation from an arbitrarily small area in a dry environment, to exceed
$PET_e$. In general, we have
$$ET \leq PET_e \leq PET_a \tag{2}$$

The original CR (Bouchet, 1963) assumes that the decrease in $ET$ equals the increase in $PET_a$,
relative to $PET_e$, from wet to dry conditions.
$$PET_e - ET = PET_a - PET_e \tag{3}$$

If the net radiation remains the same from wet to dry conditions, the decrease in latent heat
therefore equals the increase in sensible heat, i.e., $\lambda PET_e - \lambda ET = H - H_w$, where $\lambda$ is the latent
heat of vaporization and $H_w$ the sensible heat under wet conditions. Considering potential
variations in the relationship between changes in $H$ and $\lambda PET_a$, it is reasonable to assume that the
left and right hand-sides of equation (3) are proportional (Szilagyi, 2007), resulting in a generalized
linear CR in the form of
$$PET_e - ET = k(PET_a - PET_e) \tag{4}$$

or
$$ET = (1 + k)PET_e - kPET_a \tag{5}$$

where $k$ is the coefficient of proportionality, and it can be interpreted as a measure of asymmetry
for the CR. Equation (4) is identical to the original CR, i.e., equation (3) with $k = 1$, otherwise the
CR becomes asymmetric, the latter has been widely supported with observations (Kahler and
Brutsaert, 2006; Szilagyi, 2007, 2021). This asymmetry is likely to have arisen from changes in
the net radiation between wet and dry conditions and/or the energy transfer from the surrounding





environment. For example, as the land surface dries up, the evaporation pan would receive more
energy from its side and bottom and local advection of energy, resulting in a larger increase in
$PET_a$ than the decrease in $ET$ would suggest (Kahler and Brutsaert, 2006).

It is worth noting that the existing CR formulations can be interpreted in the generalized form with
equation (5), and the coefficient of proportionality $k$ for each formulation is shown in Table 1.
However, as the physical meaning of the coefficient $k$ remains unknown and unclear, it cannot be
directly estimated and need to be calibrated, which limits practical application of the
complementary relationship. Here we derive the complementary relationship with a new
expression for the coefficient of proportionality, $k$, that has a clearer physical interpretation.

**Table 1.** The coefficient of proportionality ($k$) for different CR formulations.

| Coefficient of proportionality ($k$) | Complementary relationship | References |
|---|---|---|
| $k = 1$ | $ET = 2PET_e - PET_a$ | (Bouchet, 1963) |
| $k = \dfrac{1}{b}$ | $ET = \left(1 + \dfrac{1}{b}\right)PET_e - \dfrac{1}{b}PET_a$ | (Brutsaert and Parlange, 1998) |
| $k = \dfrac{\gamma}{\Delta}$ | $ET = \left(1 + \dfrac{\gamma}{\Delta}\right)PET_e - \dfrac{\gamma}{\Delta}PET_a$ | (Granger, 1989; Szilagyi, 2007) |
| $k \approx 0.22$ | $ET = \left(\dfrac{PET_e}{PET_a}\right)^2 (2PET_e - PET_a)$ | (Brutsaert, 2015) |
| $k = \dfrac{X_{min}}{1 - X_{min}}$ | $ET = \left(1 + \dfrac{X_{min}}{1 - X_{min}}\right)PET_e - \dfrac{X_{min}}{1 - X_{min}}PET_a$ | (Crago et al., 2016) |





| $k = \beta_w$ | $ET = (1 + \beta_w)PET_e - \beta_w PET_a$ | This study |


### 3. A physically-based complementary relationship

### 3.1. Estimation of ET, PETe, and PETa

As ET is controlled by the supply of energy and aerodynamic conditions, two approaches can be
used to estimate ET, i.e., the energy-based $ET_e$ and aerodynamics-based $ET_a$ (Chow et al., 1988).
The first approach is based on the surface energy balance, i.e., partitioning of the net radiation
between sensible ($H$) and latent ($\lambda ET_e$) heat, and the ratio between the two ($H/\lambda ET_e$) is the Bowen
ratio ($\beta$). Therefore, the latent and sensible heat for a given area can be expressed as

$$\lambda ET_e = \frac{R_n}{1 + \beta} \tag{6}$$


$$H = \frac{\beta R_n}{1 + \beta} \tag{7}$$


where $R_n$ ($J \cdot m^{-2} \cdot s^{-1}$) is the net radiation minus ground heat flux (hereafter termed the net
radiation for simplicity) and equals the sum of latent and sensible heat.

The second approach is based on the aerodynamics, i.e., transport of water vapor and sensible heat
away from the evaporative surface, and the latent ($ET_a$) and sensible ($H$) heat are given by

$$\lambda ET_a = \frac{\rho c_p (e_s - e_a)}{\gamma r_a} \tag{8}$$


$$H = \frac{\rho c_p (T_s - T_a)}{r_a} \tag{9}$$


where $\rho$ is the air density ($kg \cdot m^{-3}$), $c_p$ the specific heat of air at constant pressure ($J \cdot kg^{-1} \cdot$
$K^{-1}$), $\gamma$ the psychrometric constant ($Pa \cdot K^{-1}$), and $r_a$ the aerodynamic resistance ($s \cdot m^{-1}$) that
depends on wind speed and land surface characteristics. The term $e_s - e_a$ is the difference in vapor





pressure ($Pa$) between the evaporative surface and the air above, and $T_s - T_a$ the difference
between surface ($T_s$) and air ($T_a$) temperatures ($K$).

Considering that the latent and sensible heat estimated from the two approaches above should be
identical, the Bowen ratio, $\beta$, in equations (6) and (7) can be estimated using equations (8) and (9)
as
$$\beta = \frac{\gamma(T_s - T_a)}{(e_s - e_a)} \qquad (10)$$


To estimate $PET_e$ and $PET_a$, we introduce a wet Bowen ratio ($\beta_w$), i.e., the Bowen ratio when the
evaporative surface is saturated (Zhou and Yu, 2024). By replacing $e_s$ in equation (10) with the
saturation vapor pressure ($e_s^*$) at the surface temperature ($T_s$), we have
$$\beta_w = \frac{\gamma(T_s - T_a)}{(e_s^* - e_a)} \qquad (11)$$


For a large area, $PET_e$ can be estimated based on a partition of the net radiation into latent ($\lambda PET_e$)
and sensible ($H_w$) heat when the whole area becomes saturated:
$$\lambda PET_e = \frac{R_n}{1 + \beta_w} \qquad (12)$$

$$H_w = \frac{\beta_w R_n}{1 + \beta_w} \qquad (13)$$


The situation becomes complicated when we consider $PET_a$, i.e., the ET that would occur from a
small, saturated area within a large dry area, such as an evaporation pan placed in a desert. The
saturated area is considered to be so small that its presence has no practical effect on the





surrounding environment where the wind speed, air temperature, and humidity are largely dictated
by the prevailing meteorological condition over the dry area. The meteorological variables, i.e.,
$e_a$, $T_a$, and $r_a$, over the small, saturated area are thus identical to those over the surrounding dry
environment. The surface temperature of the small, saturated area approaches its maximum value,
i.e., $T_s$ of the surrounding dry area, sustained by heat transfer from the surrounding environment.
This implies that the sensible heat of the locally saturated area is maximized and equal to that of
the large dry area given in equation (9). For the small, saturated area, the wet Bowen ratio, $\beta_w$, can
be estimated using equation (11) and meteorological variables over the dry area. Consequently,
$PET_a$ can be estimated by replacing $ET_a$ with $PET_a$, and $e_s$ with $e_s^*$ in equation (8):
$$\lambda PET_a = \frac{\rho c_p (e_s^* - e_a)}{\gamma r_a} \tag{14}$$

Equation (14) can be further simplified noting the definition of the sensible heat ($H$, equation (9))
and that of the wet Bowen ratio ($\beta_w$, equation (11)):
$$\lambda PET_a = \frac{H}{\beta_w} \tag{15}$$

As the small wet area has the same surface temperature and atmospheric conditions as the
surrounding large dry area, with the only difference being that the small area is saturated with
saturation vapor pressure at its surface ($e_s^*$), evaporation from this small wet area, i.e., $PET_a$,
represents the evaporative demand imposed by atmospheric humidity and aerodynamic conditions
(Fig. 1a). Since $PET_a$ is not constrained by available energy and unrelated to land-atmosphere
feedbacks, it cannot be realized over a large area where energy supply is limited and surface
moisture can significantly impact the atmosphere. However, $PET_a$ could be measured with a small
evaporation pan while maintaining its surface temperature equal to that of the surrounding
environment ($T_s$). Alternatively, $PET_a$ can be estimated using $H$ and $\beta_w$ from eddy covariance and





meteorological measurements, following equation (15), or using a modified Penman equation
based on routine meteorological observations (see Section 4.4) (Zhou and Yu, 2024). In contrast,
$PET_e$ represents the maximum ET that would occur over the large area, for the same amount of
net radiation but with unlimited water supply (Fig. 1b). $PET_e$ cannot be directly observed unless
the entire surface is saturated, such as over a lake or the ocean, but it can be calculated using
equation (12) (Zhou and Yu, 2024, 2025).

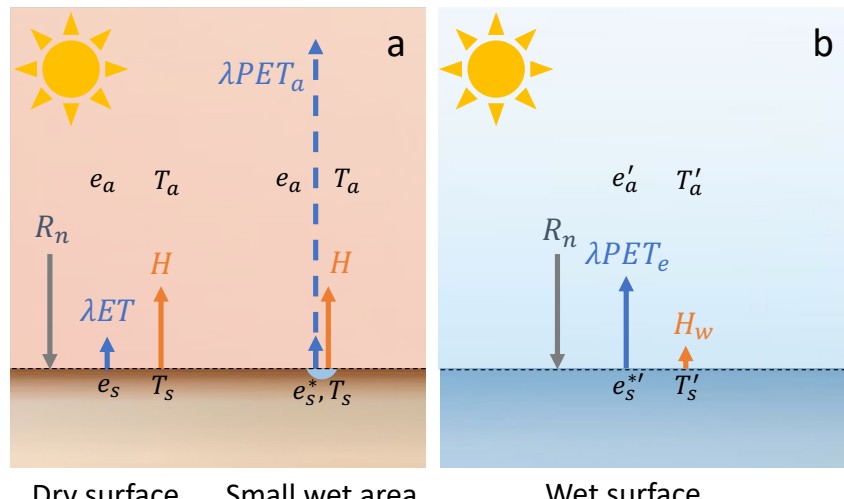


**Figure 1.** Illustration of the relationships between actual ET and two potential ET (PETe and PETa)

under wet and dry conditions. (a) In a dry environment, the net radiation ($R_n$) is partitioned into

latent heat ($\lambda ET$) and sensible heat ($H$). For an arbitrarily small wet area, the surface temperature

($T_s$), air temperature ($T_a$), and vapor pressure ($e_a$) remain the same as the surrounding dry

environment. The only difference is the vapor pressure at the evaporative surface ($e_s$ versus $e_s^*$).

Consequently, $H$ over the small wet area remains unchanged compared to the surrounding

environment, but $\lambda PET_a$ is much larger than $\lambda ET$, as both water and energy supply are not limiting



over the small wet area. (b) When the entire area becomes saturated, both surface and atmospheric
conditions cool down ($T_s'$ and $T_a'$) and become more humid ($e_s^{*\prime}$ and $e_a'$) relative to the dry
environment. As a result, latent heat ($\lambda PET_e$) increases while sensible heat ($H_w$) decreases,
constrained by $R_n$.

Considering coupled changes in temperature and humidity at the evaporative surface and of the
air, it has been demonstrated that $\beta_w$ estimated from meteorological variables corresponding to the
large dry area remains relatively constant when the whole surface becomes saturated, provided
that the net radiation remains the same under wet and dry conditions (Zhou and Yu, 2024). It is
therefore reasonable to assume that $\beta_w$ in equation (15) for a small, saturated area within a large
dry area is the same as that in equation (12) for the large area when the whole area is saturated.

**3.2. Derivation of a physically-based complementary relationship**
For a certain large area, the energy balance equation is given as
$$R_n = \lambda ET + H \tag{16}$$

The sensible heat, $H$, is directly related to $PET_a$ as shown in equation (15). Considering that $R_n$
for the given area is assumed to be the same when the whole area becomes saturated, i.e., $\lambda ET +$
$H = \lambda PET_e + H_w$, the only difference is how $R_n$ is partitioned into latent and sensible heat. With
$R_n$ from equation (12) and $H$ from equation (15), the energy balance equation can be re-written as:
$$ET = (1 + \beta_w)PET_e - \beta_w PET_a \tag{17}$$

The structural similarity between equations (5) and (17) suggests that the coefficient of
proportionality $k$ equals the wet Bowen ratio $\beta_w$. Equation (17) clarifies the physical basis for the
complementary relationship, and suggests that the actual $ET$ is a linear combination of $PET_e$ and





$PET_a$ for both wet and dry environments. This is consistent with the boundary condition for
equation (2), i.e., $ET = PET_e = PET_a$ for saturated surfaces and $ET < PET_e < PET_a$ for
unsaturated surfaces. The complementary relationship emerges as the land surface dries up, $ET$
drops below its potential, $PET_e$, and the energy not partitioned to $\lambda ET$ is shifted to increase the
sensible heat, resulting in a higher $PET_a$, relative to $PET_e$. The complementary relationship
between $ET$ and $PET_a$ essentially reflects the shift in the partitioning of the net radiation between
latent and sensible heat under different environmental conditions. In addition, we note $ET$ is
linearly related to $PET_e$, which represents the maximum $ET$ that would occur given the net
radiative energy supply. This indicates that $PET_e$ represents the atmospheric evaporative demand
or the energy constraint that controls and drives $ET$ over land, and $PET_a$, on the other hand, is in
fact a response of the atmosphere to the reduction in $ET$ where water supply is limited at the land
surface.

**3.3. Validation of the complementary relationship**
The Fluxnet2015 dataset, which provides meteorological measurements and observed land-
atmosphere exchanges of water and energy fluxes based on the eddy covariance technique from
212 sites (>1500 site-years) around the globe (Pastorello, 2020), were used to validate the
physically-based complementary relationship in equation (17). These sites cover a wide range of
climate conditions and vegetation types and were used to examine the relationship between the
actual ET and potential ET, i.e., $PET_e$ and $PET_a$. Data were included in this analysis for site-years
where measured or high-quality gap-filled data of air temperature, surface soil temperature,
sensible and latent heat fluxes were available. To reduce uncertainties in ET measurements, days
with air temperature less than 5 ℃ or negative sensible/latent fluxes were excluded. Finally, we





selected 146 Fluxnet sites with effective records of more than 90 days (see Table S1). To validate
the complementary relationship for different sites and different seasons, we used data from 7352
site-months, with effective records of more than 15 days for each month. For each site-month, the
net radiation minus ground heat flux ($R_n$) was calculated as the sum of latent and sensible heat.
$\beta_w$ was estimated from equation (11). $PET_e$ and $PET_a$ were estimated from equations (12) and
(15), respectively.

To illustrate the complementary relationship between $ET$ and $PET_a$ and the proportional
relationship between $ET$ and $PET_e$, equation (17) can be scaled and re-written as
$$\frac{ET}{PET_e} = (1 + \beta_w) - \beta_w \frac{PET_a}{PET_e} \tag{18}$$

$$\frac{ET}{PET_a} = (1 + \beta_w)\frac{PET_e}{PET_a} - \beta_w \tag{19}$$

Estimation of the three types of ET and their relationships are shown in Table 2 and Fig. 2a-c.
Based on observations from the 146 Fluxnet sites (7352 site-months), the complementary
relationship between $ET$ and $PET_a$ and the proportional relationship between $ET$ and $PET_e$ across
wide-ranging climate conditions are clearly evident (Fig. 2d-f). The scaled ET with $PET_e$ increases
from 0 to 1 and the scaled $PET_a$ decreases from 18 to 1 from the driest to the wettest site-months.
This provides observational evidence for the complementary principle that $ET$ and $PET_a$ converge
towards $PET_e$ and they closely match each other ($ET = PET_e = PET_a$) under wet conditions,
while $ET$ falls below $PET_e$ and $PET_a$ rises above $PET_e$ ($ET < PET_e < PET_a$) in a dry
environment. As implied by equation (18), a strong negative correlation (r = -0.86) was found
between the scaled $ET$ and $PET_a$, with a low level of nonlinearity induced by variations in $\beta_w$
(0.05-0.25) across the 7352 site-months (Fig. 2e). The scaled $ET$ and $PET_e$ with $PET_a$ in equation



(19), on the other hand, are positively correlated (r = 0.99, Fig. 2f), indicating the strong positive
control of energy-based $PET_e$ on $ET$ across a wide range of climate environment.

**Table 2.** Estimation of the three types of ET and their relationships.

| Equation | Range |
|---|---|
| $\lambda ET = \dfrac{R_n}{1+\beta}$ | $\left[0, \dfrac{R_n}{1+\beta_w}\right]$ |
| $\lambda PET_e = \dfrac{R_n}{1+\beta_w}$ | $\dfrac{R_n}{1+\beta_w}$ |
| $\lambda PET_a = \dfrac{H}{\beta_w}$ | $\left[\dfrac{R_n}{1+\beta_w}, \dfrac{R_n}{\beta_w}\right]$ |
| $\dfrac{ET}{PET_e} = \dfrac{1+\beta_w}{1+\beta}$ | $[0,1]$ |
| $\dfrac{ET}{PET_a} = \dfrac{\beta_w}{\beta}$ | $[0,1]$ |
| $\dfrac{PET_e}{PET_a} = \dfrac{1+\dfrac{1}{\beta}}{1+\dfrac{1}{\beta_w}}$ | $\left[\dfrac{\beta_w}{1+\beta_w}, 1\right]$ |












**Figure 2.** The complementary relationship between ET and PETa and the positive relationship
between ET and PETe. (a-c) Relationships among ET, PETe and PETa with a constant value of $\beta_w$
ranging from 0.05 to 0.25 at an increment of 0.05. (d-f) Relationships among monthly ET, PETe
and PETa estimated using meteorological and flux measurements from the Fluxnet2015 dataset
(146 sites and 7352 site-months in total, see Table S1). $\beta_w$ is shown as variations in color for each
site-month (Fig. 2e,f). Since $\beta_w$ ranged from 0.05 to 0.25 across the 7352 site-months, the slope
($\beta_w$) would vary by a factor of 5 in Fig. 2e, while for Fig. 2f the range in slope ($1+\beta_w$) varies by
19% only.

**4. Discussion**
**4.1. Assumptions for the physically-based complementary relationship**
Assumptions underlying the CR formulation in equation (17) include that 1) the net radiation, $R_n$,
for a given area remains the same under wet and dry conditions; 2) the wet Bowen ratio, $\beta_w$,
remains the same for the large area when it becomes saturated and for the small, saturated area
within the large dry area, i.e.,

$$\beta_w = \frac{H}{\lambda PET_a} = \frac{H_w}{\lambda PET_e} \qquad (20)$$

and 3) the surface temperature for the small, wet area is the same as the surface temperature for
the large dry area. The first assumption is consistent with that of the original CR and has been
adopted for estimating the $PET_e$, as we do not know in practice what the net radiation would be
when the surface becomes saturated (Zhou and Yu, 2024). Even if we forego the assumption of an
invariant net radiation and allow the net radiation to change with the surface moisture content, the
complementary relationship still holds so long as the latent heat and sensible heat change in
opposite directions (Appendix A). In addition, when scaled with $PET_e$, i.e., the energy constraint,



a clear complementary relationship between $ET$ and $PET_a$ emerges across different
regions/months, as shown in Fig. 2.

The second assumption differs from the assumption of the original CR (equation (3)) in which the
increase in $\lambda PET_a$ equals the increase in the sensible heat. The original CR, however, is not
supported by observations (Kahler and Brutsaert, 2006; Szilagyi, 2007). In contrast, the
assumption underpinning equation (20) has been shown to be reasonable as the difference in $\beta_w$
under the wet and dry conditions is very small due to coupled changes in temperature and humidity
between the land surface and the atmosphere (Zhou and Yu, 2024). Considering the potential
variations in $\beta_w$ under wet and dry conditions, we further demonstrate that the complementary
relationship between $ET$ and $PET_a$ and the positive relationship between $ET$ and $PET_e$ remain
valid on theoretical grounds (Appendix A).

Equation (20) can be re-written to show that the relative increase, rather than the absolute increase,
in the $PET_a$ and sensible heat are identical under wet and dry conditions:
$$\frac{PET_a - PET_e}{PET_e} = \frac{H - H_w}{H_w} \tag{21}$$

This is because the net radiation not expended on $ET$ is shifted to augment $H$ as the land surface
dries up. This causes $PET_a$ to increase by a factor of $1/\beta_w$, i.e., the ratio of latent over sensible
heat for a saturated surface in equation (15). This provides an explanation of the observed
asymmetric relationship between $ET$ and $PET_a$ and clarifies the physical meaning of the
coefficient of proportionality in the generalized linear form of the complementary relationship, i.e.,
$k = \beta_w$. In essence, $\beta_w$ is the source and a measure of the degree of asymmetry in the
complementary relationship.






The third assumption of identical surface temperature of the small wet area and large dry area was
invoked in order to calculate the saturation vapor pressure at the wet surface, hence the wet Bowen
ratio $\beta_w$ with equation (11). The surface temperature of the small wet area, such as a Class A
evaporation pan, would likely be lower than the surface temperature of the surrounding dry area,
so would the associated saturation vapor pressure. Thus, equation (15) is best seen to yield an
upper limit for $PET_a$ in practice. A lower $PET_a$ based on pan evaporation measurements would
increase the empirically fitted parameter $k$ in equation (4), which becomes closer to unity and
makes the observed CR less asymmetric. At the same time, the Bowen ratio of the evaporation pan
would be lower than $\beta_w$ when its surface temperature is lower than $T_s$ of the surrounding dry area.
This is because $\beta_w$ is positively related to $T_s$, as shown by rewriting equation (11) as follows:
$$\beta_w = \frac{\gamma}{\Delta + \dfrac{e_a^* - e_a}{T_s - T_a}} \tag{22}$$

where $\Delta$ is the slope of the saturation vapor pressure-temperature curve evaluated between $T_s$ and
$T_a$. Thus, the coefficient of proportionality, $k$, estimated from observations would diverge from the
Bowen ratio of an evaporation pan, and they converge to and give physical meaning to the
parameter $k$, i.e., $k = \beta_w$, only when the surface temperature of the pan is the same as its
surrounding environment. This explains why previous CRs as well as their parameters remain
empirical (Table 2).

**4.2. Interpretation of the physically-based complementary relationship**
The new derivation of the CR, i.e., equation (17), based on physical reasoning confirms that the
complementary relationship is fundamentally governed by how energy is partitioned between
latent and sensible heat under wet and dry conditions. Changes in the partitioning between the





latent and sensible heat manifest themselves as a complementary relationship between $ET$ and
$PET_a$, as the latent heat is directly related to $ET$ and the sensible heat proportional to $PET_a$ with
the wet Bowen ratio, $\beta_w$, as the coefficient of proportionality. While this physical mechanism is
broadly consistent with the concept of the original CR and the existing CR formulations (Bouchet,
1963; Brutsaert and Parlange, 1998; Brutsaert, 2015; Crago et al., 2016; Granger, 1989; Szilagyi,
2007; Szilagyi et al., 2017; Tu et al., 2023), the physically-based CR formulation clarifies the
nature of the complementary relationship between $ET$ and $PET_a$ and provides new insight into the
physical basis for the complementary relationship.

The theoretically sound complementary relationship also merits further interpretation of the
relationship between $PET_e$ and $PET_a$. Equation (17) can be re-arranged as:
$$PET_a = \left(1 + \frac{1}{\beta_w}\right) PET_e - \frac{1}{\beta_w} ET \qquad\qquad (22)$$
$PET_a$ that is related to the atmospheric condition can be seen to depend on the energy constrained
$PET_e$ and moisture constrained $ET$. Let us consider two extreme cases: 1) where $ET = PET_e$ for
saturated areas, such as the ocean and large lakes, we have the minimum of $PET_a$ as $PET_e$; 2)
where $ET = 0$ for surfaces of maximum dryness without supply of water vapor to the air, we have
the maximum of $PET_a$ as $\left(1 + \frac{1}{\beta_w}\right) PET_e$ and the largest difference between $PET_a$ and $PET_e$, i.e.,
$\frac{PET_e}{\beta_w}$. This interpretation reinforces the notion that estimation of the $PET_a$ based on the
aerodynamics responds to water vapor supply via $ET$ in addition to the energy constraint. The drier
the air with reduced $ET$, the greater the difference between $PET_a$ and $PET_e$.

**4.3. Comparison with previous formulations of the complementary relationship**



Table 1 shows that most of the existing CR formulations share the same structure as equation (17).
This is because these CR formulations are developed based on the same physical principles, i.e.,
partitioning of the net radiation shifting from latent to sensible heat from wet to dry conditions,
from which the complementary relationship originates (Bouchet, 1963). The physically-based CR
is identical to the two empirical CRs, each involving an empirical parameter, and can be used to
interpret and give meaning to these parameters. For the asymmetric CR of Brutsaert and Parlange
(1998), we have $b = \frac{1}{\beta_w}$, and $X_{min} = \frac{\beta_w}{1+\beta_w}$ for the rescaled CR of Crago et al. (2016). In particular,
the physical meaning of $X_{min}$ in the rescaled CR, i.e., the minimum value of $\frac{PET_e}{PET_a}$ when $ET$ reaches
zero, is consistent with our estimation of $PET_e$ and $PET_a$ (Table 2).

For the existing CR formulations without any parameters, or the so-called calibration-free
formulations, they are essentially special cases of the physically-based CR. For example, when the
air is saturated ($e_a = e_a^*$), we have $\beta_w = \frac{\gamma(T_s - T_a)}{(e_s^* - e_a^*)} = \frac{\gamma}{\Delta}$, the CR of Szilagyi (2007) becomes identical
to the physically-based CR. However, this special case rarely occurs, even for saturated surfaces
like oceans and lakes.

The non-linear CR formulation is also similar to the physically-based CR when $\beta_w \approx 0.22$
(Brutsaert, 2015). While this non-linear CR formulation has been validated with experimental data,
it is not realistic, however, under the driest conditions, i.e., $\frac{PET_e}{PET_a} < \frac{\beta_w}{1+\beta_w}$. This is because the
boundary condition of the non-linear CR, i.e., $\frac{PET_e}{PET_a} \to 0$ when $ET \to 0$, is not physically sound, as
$PET_a$ cannot be infinitely large and in fact $PET_a$ is limited by $\frac{R_n}{\beta_w}$ as $ET$ reaches zero and $H$ equals



$R_n$ (Table 2). To overcome this problem, the non-linear CR has been combined with the rescaled
CR to develop yet another calibration-free CR (Szilagyi et al., 2017). However, this formulation
still represents a special case without physically meaningful parameters such as $\beta_w$ to account for
variations in the complementary relationship under different conditions.

Compared with these earlier formulations, the physically-based CR has many distinct advantages.
First, $PET_e$ and $PET_a$ are clearly defined based on physical processes and can be estimated using
observed data. Second, the basis for this new CR is physically sound and its derivation is purely
based on shifts in the surface energy balance under wet and dry conditions with three assumptions.
Third, the physical meaning of the coefficient of proportionality, i.e., the wet Bowen ratio $\beta_w$, is
clear and it accounts for the degree of asymmetry in the complementary relationship across a wide
range of environmental conditions. Moreover, $\beta_w$ can be directly estimated from observed data
without any calibration (equation (11)). This physically-based CR can therefore be widely applied
for estimating $ET$ across different regions at various time scales. Finally, the physically-based CR
clearly quantifies the complementary relationship between $ET$ and $PET_a$ and the positive
relationship between $ET$ and $PET_e$, i.e., equations (18) and (19). This provides an enhanced
understanding of the relationships among the three types of ET over land.

**4.4. Implications for practical applications of the complementary relationship**
Application of the physically-based CR for ET estimation requires values for $\beta_w$, $PET_e$ and $PET_a$,
all of which can be calculated using observations of meteorological variables and surface
temperature. However, the fact that surface temperature data are not readily available may restrict



broader application of this method for ET estimation. To address this limitation, the wet Bowen
ratio, $\beta_w$, can be approximated as a function of air temperature:
$$\beta_w = \alpha \cdot \frac{\gamma}{\Delta} \tag{23}$$

where $\Delta$ is the slope of the curve for saturation vapor pressure as a function of temperature ($Pa \cdot$
$K^{-1}$). The coefficient $\alpha$ typically varies from 0.15 to 0.3, and an approximate value of $\alpha \approx 0.24$
can be adopted (Yang and Roderick, 2019; Zhou and Yu, 2024).

While $PET_e$ and $PET_a$ are defined based on energy balance and aerodynamic principles, they can
also be derived using modified versions of the Penman equation with an adjustment parameter $k'$
(Zhou and Yu, 2024). When the evaporative surface is saturated, the Penman equation can be used
to estimate both $PET_e$ and $PET_a$, i.e., $ET = PET_e = PET_a$ (Penman, 1948). However, the direct
application of the Penman equation would overestimate $PET_e$ (Milly and Dunne, 2016, 2017;
Zhou and Yu, 2025) and simultaneously underestimate $PET_a$ in a dry environment. This occurs
because $PET_e$ assumes a large, saturated surface (Fig. 1b), leading to an overestimation of its
aerodynamic component when observed meteorological variables corresponding to dry surface
conditions are used by the Penmen equation. Concurrently, the energy supply required to sustain
$PET_a$ for a small, saturated surface is underestimated (Fig. 1a), as the Penman equation does not
account for energy transferred from the surrounding dry environment. These issues can be resolved
with the adjustment parameter $k'$ (Zhou and Yu, 2024).
$$\lambda PET_e = \frac{\Delta R_n + k' \dfrac{\rho c_p (e_a^* - e_a)}{r_a}}{\Delta + \gamma} \tag{24}$$



$$\lambda PET_a = \frac{\Delta R_n/k' + \frac{\rho c_p(e_a^* - e_a)}{r_a}}{\Delta + \gamma} \qquad (25)$$

where $e_a^*$ is the saturation vapor pressure at air temperature ($Pa$), and $e_a^* - e_a$ is the vapor pressure
deficit. The parameter $k'$ can be determined by equating $PET_e$ from equations (12) and (24), and
$k'$ so determined can be used to compute $PET_a$ with equation (25). Thus, $\beta_w$, $PET_e$ and $PET_a$ can
be estimated using routine meteorological variables, enabling broader applications of the
physically-based CR for ET estimation.

**5. Conclusions**
In this study, we proposed definitive estimators of the $PET_e$ and $PET_a$ based on the energy balance
and aerodynamic principles and derived an alternative complementary relationship with a
physically meaningful parameter, i.e., the wet Bowen ratio, which can be directly calculated from
observations. The complementary relationship between $ET$ and $PET_a$ fundamentally originates
from partitioning of the net radiation between latent and sensible heat, with $ET$ directly related to
the latent heat and $PET_a$ proportional to the sensible heat. The wet Bowen ratio for a small,
saturated area quantifies the degree of asymmetry in the complementary relationship. In addition,
$ET$ is linearly related to $PET_e$, with the latter representing the evaporative demand of the
atmosphere or the energy constraint on $ET$ over land. By clarifying the quantitative relationship
between the three types of $ET$, this study advances our understanding of the complementary
relationship and would promote and facilitate practical application of the complementary
relationship for ET estimation across different regions and time scales.



## Appendix A: A variant of the complementary relationship for evapotranspiration

To derive the complementary relationship in equation (17), it has been assumed that 1) the net radiation for a dry area remains the same as the entire area becomes saturated; 2) the wet Bowen ratio for a small, saturated area within a large dry area is the same as that over the entire area if it were saturated. Here we consider and allow potential variations in the net radiation and the wet Bowen ratio under wet and dry conditions and re-examine the effectiveness of the complementary relationship between ET and $PET_a$ and the nature of the relationship between ET and $PET_e$.

For the first assumption, let $R_n$ be the observed net radiation for the dry area, and the net radiation for the entire area if it were saturated is $R_n/k_1$ with $k_1 > 0$. Similar to the energy balance for the dry area shown in equation (16), the energy balance for the area when saturated is given by

$$R_n/k_1 = \lambda PET_e + H_w \tag{A1}$$

For the second assumption, let $\beta_w$ be the wet Bowen ratio over the small, saturated surface, and the Bowen ratio if the entire area were saturated is $k_2\beta_w$ with $k_2 > 0$. The sensible heat for the large saturated area, i.e., $H_w$, is expressed as

$$H_w = k_2\beta_w \lambda PET_e \tag{A2}$$

Equation (A1) can be re-written as:

$$R_n/k_1 = (1 + k_2\beta_w)\lambda PET_e \tag{A3}$$

Replacing $R_n$ with $\lambda ET + H$, and noting $H$ equals $\beta_w \lambda PET_a$, we have

$$(\lambda ET + \beta_w \lambda PET_a)/k_1 = (1 + k_2\beta_w)\lambda PET_e \tag{A4}$$

or

$$ET = k_1(1 + k_2\beta_w)PET_e - \beta_w PET_a \tag{A5}$$



Equation (A5) suggests that the complementary relationship between $ET$ and $PET_a$ and the
proportional relationship between $ET$ and $PET_e$ still hold despite the potential differences in the
net radiation and the wet Bowen ratio under wet and dry conditions. It is clear that equation (A5)
is reduced to equation (17) when $k_1 = k_2 = 1$. Therefore, equation (A5) represents a more
generalized complementary relationship without the restrictive assumptions required by equation

481    (17).


**Data availability.** The Fluxnet2015 dataset is publicly available from
https://fluxnet.org/data/fluxnet2015-dataset/.

**Author contribution.** S.Z. conceived the study, performed the data analysis, and wrote the initial
manuscript. B.Y. provided critical revisions and edits.

**Competing interests.** The authors declare that they have no conflict of interest.

**Acknowledgements.** We acknowledge all the principal investigators who contributed data to the
Fluxnet2015 dataset (Table S1). This work was supported by the National Key Research and
Development Program of China (2022YFF0801303), National Natural Science Foundation of
China (42471108), and the Fundamental Research Funds for the Central Universities.

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
