# Peer review of "New derivation and interpretation of the complementary relationship for evapotranspiration 3 Sha Zhou1,2\*, Bofu Yu3 4 1State Key Laboratory of Earth Surface Processes and Hazards Risk Governance (ESPHR), 5 Faculty of Geographical Science, Beijing Normal Universi"

_EGUsphere, 2025_

## Author Comment (AC1)

**The authors claim to have derived a new physically-based CR via 'rigorous derivation based on physics', unlike previous versions, which they deem only empirical.**

In fact, what the authors achieve is making use of several hypothetical and highly speculative assumptions (lines 210-215):

i) The surface temperature of a small, freely evaporating water body is always the same as that of the surrounding drying land (this would require a heat conduction as effective as evaporative cooling, which is highly unlikely under realistic conditions, thus the corresponding potential evaporation rate remains speculative only);

ii) The Bowen ratio (βw) written for such a small water body does not change during drying of the environment (contradicting the constant surface net radiation assumption stated).

None of the above assumptions are valid in general and none been ever confirmed rigorously by any study.

Response:
We respectfully disagree with the reviewer's comment. Assumption i) is essential for formulating a complementary relationship with a physically meaningful parameter, while Assumption ii) has been validated in our previous studies (Zhou and Yu, 2024 Journal of Hydrology; Zhou and Yu, 2025 Global Change Biology).

To derive a complementary relationship (CR) for evapotranspiration (ET), we must:
1) Define and estimate potential ET (PET) and apparent potential ET (PETa).
2) Establish the relationship among ET, PET, and PETa.

Definition of PET and PETa:
PET represents the rate of ET that would occur when water supply is unlimited at the evaporative surface (i.e., a fully saturated surface). Since PET is constrained by available energy (the net radiation), it is estimated using the energy-based approach and termed as PETe in our study. PET cannot be directly observed unless the entire surface is saturated, such as over a lake or the ocean. Our comparison of PET equations, including Priestley-Taylor and Penman equations, demonstrates that PETe, estimated using net radiation and the wet Bowen ratio ($\beta_w$), is the most reliable PET estimator (Zhou and Yu, 2024; 2025).

PETa represents the ET rate from a small, saturated surface within a larger, unsaturated area (e.g., an evaporation pan), with energy supplied by both net radiation and the surrounding environment. However, the energy transfer from the surroundings varies with the pan size, leading to variations in the surface temperature of the wet area and the corresponding evaporation rate. Due to the inherent ambiguity in the definition of PETa, it is essentially indeterminate in practice. The lack of a definitive estimator further complicates PETa estimation, making it challenging to develop a physically-based CR formulation. To address this issue, we estimate the upper limit for PETa by assuming that the surface temperature of the small wet area is maximized and equal to that of the surrounding dry area (Ts).

Complementary Relationship Framework:
The complementary relationship in essence describes the interplay between ET, PET, and PETa. While their exact quantitative relationships remain uncertain across different formulations (Table S2), all valid CR formulations must satisfy two boundary conditions:
- ET = PET = PETa under wet conditions.
- ET < PET < PETa as the surface dries up.

Comparison with Previous CR formulations:
We derived the CR formulation using two well-defined estimators of PET and PETa, along with a physically meaningful parameter ($\beta_w$):
- Assumption ii) is used to estimate PET, i.e., PETe.
- Assumption i) is used to estimate the upper limit of PETa by maintaining the surface temperature of the small wet area equal to that of the surrounding environment.

Quite dissimilar to our approach, many previous CR studies in fact involve other two assumptions, i.e., Assumptions a) and b) below. These assumptions are so commonly made implicitly that they have been taken for granted. In fact, they have rarely been clearly stated and validated.
- Assumption a): PETa can be estimated using Penman equation (PETpm) or pan evaporation.
- Assumption b): PET can be estimated using the Priestley-Taylor equation (PETpt).

We adopt Assumptions (i) and (ii) instead of Assumptions (a) and (b) for three main reasons:
1) **Reliability of PETe**: Assumption (ii) provides a robust PET estimator, i.e., PETe. Both Assumption ii) and the reliability of PETe have been validated in our previous studies (Zhou and Yu, 2024; 2025). Importantly, these studies have demonstrated that PETe is

much better than PETpt in terms of estimating ET under wet conditions (e.g., over the ocean) and that PETe can be used for estimating the PET over land based on the Budyko framework.

2) **Indeterminacy of PETa**: PETa is indeterminate, as its value varies with the size of the small wet area. Additionally, the Penman equation should not be used to estimate PETa as it neglects energy transfer from the surrounding environment (Zhou and Yu, 2024). We define an upper limit of PETa using Assumption i) to ensure a physically meaningful CR formulation. As discussed in Section 4.1, only when PETa is maximized can we derive the CR with a physical meaningful parameter ($\beta_w$). This is because the empirical parameter, $k$, estimated through calibration with observations would diverge from the Bowen ratio of an evaporation pan when its temperature is lower than the surface temperature of the surrounding environment (Ts), and they converge to and give physical meaning to the parameter $k$, i.e., $k = \beta_w$, only when the surface temperature of the pan is the same as its surrounding environment. This resolves the empirical nature of many previous CR formulations (Table 2).

3) **Consistency with the boundary conditions**: Our estimates of PETe and PETa allow CR derivation from the fundamental energy balance equation (Rn=ET+H). They also ensure consistency with two key boundary relationships, i.e., ET=PETe=PETa under wet conditions and ET<PETe<PETa when the surface dries up (see Table 2 and Fig. 2). In contrast, previous CR formulations often violate these conditions, particularly when PETpt and PETpm are directly adopted to estimate PET and PETa, respectively. For instance, using PETpt and PETpm directly results in inconsistencies under wet conditions, i.e., ET≠PETpt and PETpt≠PETpm (see Fig. S3 of Zhou and Yu, 2025 and Fig. 5 of Yang et al., 2019). Since PETpt and PETpm fail to meet the required boundary conditions, they should not be used to formulate the CR.

They proceed further and claim that neither the Penman nor the Priestley-Taylor equation is appropriate for estimating the corresponding apparent potential evaporation rate or the evaporation rate of the wet environment, even though that these equations are the backbone of practically all existing CR methods. Yet, when they decide to discuss the practical applications of their version of the CR they turn to a modified version of the Penman equation with an **empirical coefficient** (k') to be determined from measurements (eqs. 25 & 26). Note that the original Penman equation does not have this additional coefficient. Also, as the land surface temperature is typically unknown in practical applications, they introduce **another empirical coefficient** (α) to convert the Bowen ratio of equilibrium evaporation into βw in eq. (24).

One would expect that when a new method is introduced then its practical predictive superiority is showcased over existing similar methods it is supposed to replace. Such a validation is completely missing here.

Response:
We agree that Penman and Priestley-Taylor equations have been extensively used in hydrology, climate, agriculture, and many other relevant fields. However, recent research has questioned their reliability in estimating PETa and PET (Milly et al., 2016; 2017; Greve et al., 2019; Zhou and Yu, 2024; 2025).

The Penman equation (PETpm) combines PETe and PETa while eliminating the term Ts, assuming that 1) the surface is saturated and 2) PET=PETe=PETa, which are only valid under wet conditions (see Section 2 of Zhou and Yu, 2024). Direct application of PETpm over unsaturated land leads to PET overestimation (due to dry atmospheric conditions, such as higher vapor pressure deficit and warmer temperatures) and PETa underestimation (as it neglects energy transfer from the surroundings). To resolve this issue, an adjustment parameter $k' = \frac{1+1/\beta}{1+1/\beta_w}$ (where $\beta$ is the Bowen ratio and $\beta_w$ is the wet Bowen ratio) can modify the Penman equation for estimating PET and PETa using routine meteorological data when Ts is unknown (Zhou and Yu, 2024).

The Priestley-Taylor equation (PETpt), a simplified version of PETpm with an empirical coefficient ($\alpha$), has commonly been used for PET estimation. However, PETpt exhibits large biases in estimating ET over the ocean (Yang et al., 2019; Zhou and Yu, 2025), making it unsuitable for PET estimation. Moreover, PETpt overestimates the sensitivity of PET to temperature, leading to an exaggerated increase in PET under warming climates (Yang et al., 2019; Zhou and Yu, 2025). These issues can be resolved by using PETe instead of PETpt.

This study derives a CR formulation based on PETe, PETa, and the physically meaningful parameter ($\beta_w$). Multiple approaches can be used to estimate PETe, PETa, and $\beta_w$, depending on data availability:
1) When Ts and sensible heat (H) are known (e.g., flux tower sites and reanalysis products), $\beta_w$ can be calculated using equation (11), while PETe and PETa can be derived from equations (12) and (15), respectively.
2) When Ts is available (from *in situ* or remote sensing observations) but H is not, PETa can also be estimated using equation (14).
3) When both Ts and H are unknown, $\beta_w$ can be estimated from routine meteorological observations using equation (23) and PETe and PETa using equations (24) and (25).

Based on these approaches to estimation of PETe, PETa and $\beta_w$ depending on data availability, the newly derived CR formulation has significant potential for estimating ET. We have discussed its advantages comparing with previous CR formulations (Table 2) in Section 4.3 and discussed its practical implications in Section 4.4. However, applying the CR formulation specifically for ET estimation is not the primary focus of this, for most part, analytical work. A comprehensive and systematic evaluation of its effectiveness for ET estimation, along with comparisons to other established ET estimation methods (such as FLUXCOM and GLEAM), is an important next step. This should be addressed in a dedicated future study to rigorously validate whether the CR formulation can offer improved performance or advantages over existing approaches.

In this study, we aim to advance our understanding of the complementary relationship by clearly defining and estimating PETe and PETa and establishing the quantitative relationships among ET, PETe, and PETa in a CR formulation with a physically meaningful parameter ($\beta_w$). **Notably, the CR formulation in equation (17) and the relationships among ET, PETe, and PETa as shown in equations (18) and (19) have been validated using data from 146 Fluxnet sites (see Fig. 2).**

The authors' main equation (eq. 17), when combined with eqs. 12 and 15 yields simply: ET = Rn – H, which is a rather trivial formulation of the energy balance equation. All the authors do is combine this energy balance equation with the definition of the Bowen ratio and express them in a way that looks like a CR equation, i.e., their eq. 17. For βw they use the actual land surface and air temperature plus vapor pressure values (i.e., eq. 11) by capitalizing on assumption ii). An additional problem is that they still need to know H unless they employ the above mentioned modified Penman equation.

So what is the new insight from the authors' 'theoretically sound' CR? I am not sure.

Response:
We respectfully disagree with the characterization of the energy balance equation as a trivial formulation. On the contrary, it plays a fundamental role in the complementary relationship, which is governed by the partitioning of energy between latent and sensible heat under wet and dry conditions (see Section 2). Changes in the partitioning between the latent and sensible heat manifest themselves as a complementary relationship between ET and PETa, as the latent heat is directly related to ET and the sensible heat proportional to PETa with the wet Bowen ratio ($\beta_w$).

**The key new insight is that by clearly defining and estimating PET and PETa, the complementary relationship naturally emerges from the energy balance equation, which serves as its foundation.** This revelation and reification eliminate the need to construct complex, non-linear relationships among ET, PET, and PETa that rely on unknown empirical parameters, as done in many previous studies (see the review by Han and Tian, 2020 HESS). In fact, CR formulations based on PETpt and PETpm fail to satisfy the boundary conditions of the complementary relationship, and many existing CR formulations are either special cases or unrealistic under certain conditions (see our response to the first comment and Section 4.3).

**Another new insight is that the physical meaning of the CR parameter $k$, identified as the wet Bowen ratio ($\beta_w$), is explicitly clarified.** $\beta_w$ accounts for the degree of asymmetry in the complementary relationship across diverse environmental conditions. Since $\beta_w$ can be directly estimated from observed data without calibration, the physically-based CR can be applied for estimating ET across different regions and time scales.

Regarding the estimation of $\beta_w$ and PETe, Assumption ii) has been validated as $\beta_w$ remains fairly constant due to coupled changes in temperature and humidity of the air and at the land surface from dry to wet conditions (Zhou and Yu, 2024). Furthermore, $\beta_w$ provides a robust estimate of the Bowen ratio for wet surfaces. In contrast, the wet Bowen ratio derived from PETpt ($\beta_{pt}$) is highly sensitive to temperature variations between wet and dry conditions and exhibits significant biases under wet conditions (Zhou and Yu, 2024; 2025).

As for the estimation of PETa, our formulation offers flexibility by providing three distinct approaches depending on data availability (see our response to the comment above). This adaptability ensures that the CR formulation remains practical and applicable even when direct measurements of sensible heat flux (H) are unavailable. Thus, the reliance on H is not a limitation but rather a feature that enhances the versatility of our approach.

Based on these observations I can only recommend **rejection** of the manuscript. A thoroughly revised version of the manuscript that is not based on highly questionable assumptions [i.e., i) and ii)] could only be publishable if the authors demonstrate its practical predictive superiority (i.e., that it indeed leads to better ET estimates when differences in the number of parameters to calibrate and input requirements are properly accounted for) over existing CR models and drops any claim that it is a 'theoretically sound' and 'rigorously derived' CR version (in opposition to other existing CR versions) as all versions of the CR today are empirical to varying degrees, if not else then for the Penman equation (with its empirically derived wind-function) they employ.

Response:

We appreciate the reviewer's critical feedback and the opportunity to further clarify the focus and contributions of our study. While we understand the concerns raised, we respectfully argue that the theoretical advancements and foundational insights presented in this paper are significant and warrant publication, even in the absence of extensive practical validation at this stage. Below, we outline the key reasons why this study is innovative, scientifically valuable, and deserving of publication:

**1) Theoretical focus and novelty**

This paper is primarily a theoretical contribution aimed at advancing the fundamental understanding of the nature of the complementary relationship for ET. Unlike previous studies that often rely on empirical formulations and assumptions, our work provides a physically-based derivation of the CR formulation. By clearly defining and estimating PET and PETa, we establish a more robust and theoretically sound foundation for the CR. This represents a significant departure from many existing approaches, which frequently depend on empirical parameters and lack clear physical justification of the relationships among ET, PET, and PETa.

**2) Clarification of the CR parameter ($\beta_w$)**

One of the key innovations of this study is the explicit identification and interpretation of the CR parameter $k$ as the wet Bowen ratio ($\beta_w$). This parameter, which accounts for the asymmetry in energy partitioning between latent and sensible heat under varying environmental conditions, is directly estimable from observational data without the need for calibration. This eliminates the need for empirical fitting, which has been a major limitation of many previous CR formulations. By grounding $k$ in physical principles, our approach enhances the generalization and applicability of the CR across diverse regions and time scales.

**3) Resolution of boundary condition issues**

Our formulation guarantees consistency with the fundamental boundary conditions required by the CR, namely ET=PET=PETa under wet conditions and ET<PET<PETa under dry conditions. This is a critical improvement over many existing CR models, which often violate these conditions, particularly when PETpt and PETpm are used to estimate PET and PETa. By addressing these inconsistencies, our work provides a more robust framework for understanding and modeling the complementary relationship.

**4) Flexibility and adaptability**

The proposed CR formulation is designed to be flexible and adaptable to different data availability scenarios. Whether surface temperature (Ts) and sensible heat (H) are known (e.g., from flux towers or reanalysis products) or must be estimated from routine meteorological observations, our approach provides multiple pathways for estimating PET, PETa, and $\beta_w$. This adaptability ensures that the method can be applied in a wide range of practical settings, even when direct measurements are unavailable.

**5) Validation and foundational insights**

While this paper is primarily theoretical, we have validated key aspects of our CR formulation using data from 146 Fluxnet sites (see Fig. 2). These results demonstrate the robustness of our approach in capturing the complementary relationship between ET and PETa, and the positive relationship between ET and PETe. Furthermore, our findings have been supported by previous studies (Zhou and Yu, 2024; 2025), which validate the stability of $\beta_w$ and the reliability of PETe as a PET estimator.

**6) Broader implications and future directions**

The theoretical advancements presented in this paper have far-reaching implications for the fields of hydrology, climatology, and environmental science. By providing a more rigorous and physically consistent framework for the CR, our work lays the groundwork for future studies to develop improved ET estimation methods. While we acknowledge that practical validation and comparison with existing models are important next steps, these efforts are beyond the scope of this theoretical paper and should be addressed in dedicated follow-up studies.

**7) Why this paper should be published**

This paper makes a significant contribution to the scientific community by addressing long-standing theoretical challenges in the formulation and interpretation of the complementary relationship. It provides a clear, physically-based framework that resolves many of the empirical shortcomings of the existing CR formulations. While practical applications and comparisons with other methods are important, they do not diminish the value of the theoretical insights presented here. Publishing this work will enable the scientific community to build upon these foundational advancements, ultimately leading to more accurate and reliable ET estimation methods.

In conclusion, we respectfully request that the paper be considered for publication based on its theoretical rigor, innovative insights, the potential to challenge current practice, and ultimately to advance the field. We believe that the original contribution of this study will inspire further research and practical applications, making it a valuable addition to the literature on evaporation.

References:

1) Greve, P., Roderick, M. L., Ukkola, A. M., & Wada, Y.: The aridity index under global warming. Environmental Research Letters, 14(12), 124006, 2019.
2) Han, S. and Tian, F.: A review of the complementary principle of evaporation: from the original linear relationship to generalized nonlinear functions, Hydrology and Earth System Sciences, 24, 2269–2285, 2020.
3) Milly, P. C. D. and Dunne, K. A.: Potential evapotranspiration and continental drying, Nature Climate Change, 6, 946–949, 2016.
4) Milly, P. C. D. and Dunne, K. A.: A Hydrologic Drying Bias in Water-Resource Impact Analyses of Anthropogenic Climate Change, JAWRA Journal of the American Water Resources Association, 53, 822–838, 2017.
5) Yang, Y. and Roderick, M. L.: Radiation, surface temperature and evaporation over wet surfaces, Q.J.R. Meteorol. Soc., 145, 1118–1129, 2019.
6) Zhou, S. and Yu, B.: Physical basis of the potential evapotranspiration and its estimation over land, Journal of Hydrology, 641, 131825, 2024.
7) Zhou, S. and Yu, B.: Reconciling the Discrepancy in Projected Global Dryland Expansion in a Warming World, Global Change Biology, 31, e70102, 2025.

---

## Author Comment (AC2)

**1. Why the Penman equation should not be directly used to estimate PET and PETa under dry (unsaturated) conditions**

For a saturated surface, the Penman equation is derived based on two key conditions:

1) **The evaporative surface is saturated**, allowing the introduction of the slope of the saturation vapor pressure curve ($\Delta$) into the Penman equation. This eliminates the dependence on surface temperature (Ts), and combination of the energy-based ET (ETe) and the aerodynamics-based ET (ETa) yields:

$$\Delta\lambda ET_e + \gamma\lambda ET_a = \Delta R_n + \frac{\rho c_p(e_a^* - e_a)}{r_a} \quad (R1)$$

2) **Under wet conditions, ET is equal to both ETe and ETa**, leading to the classic Penman equation:

$$\lambda ET = \frac{\Delta R_n + \frac{\rho c_p(e_a^* - e_a)}{r_a}}{\Delta + \gamma} \quad (R2)$$

However, for a dry (unsaturated) surface, these conditions no longer hold. Assuming the surface is saturated, we can estimate an energy-based PET (PETe) and an aerodynamics-based PET (PETa), but under dry conditions, PETe is lower than PETa:

$$\frac{PET_e}{PET_a} = \frac{1 + \frac{1}{\beta}}{1 + \frac{1}{\beta_w}} \quad (R3)$$

where $\beta$ is the Bowen ratio and $\beta_w$ is the wet Bowen ratio. Under wet conditions, $\beta = \beta_w$ and $PET_e = PET_a$. However, under dry conditions, $\beta > \beta_w$ leading to $PET_e < PET_a$.

To account for this discrepancy, we introduce an adjustment parameter

$$k' = \frac{1 + \frac{1}{\beta}}{1 + \frac{1}{\beta_w}} \quad (R4)$$

This allows us to modify the Penman equation as follows:

$$\lambda PET_e = \frac{\Delta R_n + k'\frac{\rho c_p(e_a^* - e_a)}{r_a}}{\Delta + \gamma} \quad (R5)$$

$$\lambda PET_a = \frac{\Delta R_n/k' + \frac{\rho c_p(e_a^* - e_a)}{r_a}}{\Delta + \gamma} \quad (R6)$$

Under dry conditions, direct application of the Penman equation (R2) with observed meteorological variables overestimates the PET (or PETe) as vapor pressure deficit ($e_a^* - e_a$) is overestimated (higher than that under wet conditions, i.e., $k'(e_a^* - e_a)$). Conversely,

equation (R2) would underestimate the apparent PET (or PETa) because it neglects additional energy input from the surrounding environment. By considering this additional energy input ($A_e$), PETa over a small wet area (e.g., an evaporation pan) can be expressed as

$$\lambda PET_a = \frac{\Delta(R_n + A_e) + \dfrac{\rho c_p(e_a^* - e_a)}{r_a}}{\Delta + \gamma}, \qquad 0 < A_e < \left(\frac{1}{k'} - 1\right) R_n \ (R7)$$

Since $A_e$ is uncertain—evaporation from a small pan differs from that of a larger pan due to variations in $A_e$ (and thus Ts)—PETa becomes indeterminate. However, an upper limit for PETa can be established when the surface temperature of the small wet area approaches its maximum value, i.e., the surface temperature (Ts) of the surrounding environment, with total energy supply to the small wet area of $R_n/k'$.

For a detailed derivation of equations (R1-R6), please refer to Sections 2 and 3 and Fig. 1 of Zhou and Yu (2024).

[Figure]

**Fig. 1.** Derivation of the Penman equation under wet conditions and the modified Penman equation under dry conditions.

**2. Why the Priestley-Taylor equation should not be used to estimate PET**

Both the Priestley-Taylor equation (PETpt) and the energy-based PETe rely on energy balance principles. However, they differ in the estimation of wet Bowen ratio ($\beta_w$ and $\beta_{pt}$):

$$\lambda PET_e = \frac{R_n}{1 + \beta_w} \quad (R8)$$

$$\beta_w = \frac{\gamma(T_s - T_a)}{(e_s^* - e_a)} \quad (R9)$$

$$\lambda PET_{pt} = \alpha \frac{\Delta R_n}{\Delta + \gamma} \quad (R10)$$

$$\beta_{pt} = \frac{1}{\alpha}\frac{\gamma}{\Delta} + \frac{1}{\alpha} - 1 \quad (R11)$$

Under wet conditions, such as over the ocean, $\beta_w$ and $PET_e$ can accurately estimate the wet Bowen ratio and ET with minimal bias. However, the Priestley-Taylor equation requires an empirical parameter $\alpha$, which varies spatially, as noted by the reviewer. Using a fixed $\alpha = 1.26$ leads to systematic biases, and the bias also varies spatially:

1) Over the tropical ocean (where Ts is higher), the wet Bowen ratio is underestimated, leading to ET overestimation.

2) Over mid- and high-latitude ocean (where Ts is lower), the wet Bowen ratio is overestimated, leading to ET underestimation.

These biases are illustrated in Fig. 5 of Yang and Roderick (2019), Fig. S3 of Zhou and Yu (2025), and Fig. 3d,e of Zhou and Yu (2024).

To correct these biases, the reviewer suggested calibrating $\alpha$ so that PETpt matches ET under wet conditions. However, $\beta_{pt}$ is highly sensitive to $\frac{\gamma}{\Delta}$ (which depends on Ts) with a sensitivity coefficient of $\frac{1}{\alpha}$. Lowering $\alpha$ over mid- and high-latitude oceans may further exaggerate this sensitivity. Increasing $\alpha$ can reduce this sensitivity and align $\frac{1}{\alpha}$ with ocean observations (~0.24), but this would simultaneously lower $\beta_{pt}$ to be much lower than the wet Bowen ratio (see the $\frac{1}{\alpha} - 1$ term in equation (R10)).

Since $\beta_{pt}$ is overly sensitive to temperature changes, the Priestley-Taylor equation is unsuitable for estimating PET under dry conditions, where temperatures are significantly higher than in wet conditions. Fig. 2 and Fig. 3f,g of Zhou and Yu (2024) show that PETpt and $\beta_{pt}$ exhibit excessive sensitivity to temperature variations driven by surface moisture changes.

**This issue highlights a fundamental limitation of the Priestley-Taylor equation in estimating ET under wet conditions and the PET under dry conditions. In particular, the Priestley-Taylor equation fails to accurately capture the sensitivity of the wet**

**Bowen ratio and hence PET to temperature variations, making it unsuitable for applications in a warming climate.**

To overcome this issue, we use $\beta_w$ and PETe, because

1) $\beta_w$ can accurately estimate the Bowen ratio for wet surfaces, allowing PETe, estimated with $\beta_w$ and $R_n$, to closely match observed/projected ET over the ocean. Please see Figs. 2 and 3 of Zhou and Yu (2024) and Fig. S3 of Zhou and Yu (2025).

2) For dry surfaces, $\beta_w$ remains a reliable estimator of the Bowen ratio when the surface becomes saturated. While $\beta_w$ is estimated using observed/projected meteorological variables under dry conditions, it remains fairly constant due to coupled changes in temperature and humidity of the air and at the land surface from dry to wet conditions. Please see Figs. 2 and 3 of Zhou and Yu (2024).

[Figure]

**FIGURE 5** Relationship between Bowen ratio ($\beta$) and surface temperature ($T_s$). Colours represent data density. The three solid curves represent equilibrium evaporation (brown), the Priestley–Taylor relationship (yellow) and the best linear fit to the WHOI OAFlux dataset (black), respectively

[Figure]

**Fig. S3 Differences between the potential and actual evaporation over ocean based on ERA5 and CMIP6.** (a-c) Difference between evaporation (ET) and potential evaporation (PET) using the energy-based PETe, the Priestley-Taylor equation (PETpt), and the open-water Penman equation (PETow) for the historical period (1980-2021) based on ERA5. (d-f) The same as a-c, but for the multi-model mean difference based on CMIP6.

[Figure]

**Fig. 2.** Illustration of the variation in the wet Bowen ratio ($\beta_w$) for an ocean grid cell from the wet to hypothetical driest states (a). The saturation vapor pressure ($e_s^*/\gamma$) is expressed as a function of temperature ($T$) as shown in the black line. The blue and brown lines represent isenthalpic processes, i.e., constant $T + e/\gamma$, for the air and land surface, respectively. $\beta_w$ and the Bowen ratio ($\beta$) can be calculated under the wet (blue), dry (orange), and driest (red) states. (b and c) Changes in three Bowen ratios ($\beta_w$, $\beta_o$, and $\beta_{pt}$) and related PETs ($PET_e$, $PET_o$, and $PET_{pt}$) with the mean of surface and air temperatures. The pink lines show one possible pathway of changes in $\beta_w$ and $PET_e$ when surface and air temperatures change in proportion from the wet to the driest states, the green lines show changes in $\beta_o$ and $PET_o$ based on equation (25), and the blue lines show changes in $\beta_{pt}$ and $PET_{pt}$ based on the Priestley-Taylor equation (26). The sensitivities of three Bowen ratios and PETs to the mean temperature, for example $d\beta_w/dT$ and $dPET_e/dT$, are calculated as the ratio of their differences over the difference in the mean temperature between the wet and driest conditions. (For interpretation of the references to color in this figure legend, the reader is referred to the web version of this article.)

[Figure]

**Fig. 3.** Variations in the wet Bowen ratio ($\beta_w$) under wet and dry conditions. (a, b) Relationship between $\beta_w$ and $\gamma/\Delta$ under the wet (a) and driest (b) conditions. The color of the scatter points indicates the sample density, with the highest density shown in orange, followed by red and blue, and the lowest density in grey. Monthly data from ocean grid cells for the period 1940–2022 (996 months) in ERA5 are used. (c) Distribution of the linear regression slope between $\beta_w$ and $\gamma/\Delta$, i.e., $\alpha$ in equation (27), under the wet and driest conditions. (d) Distribution of the differences between the three Bowen ratios ($\beta_w$, $\beta_o$, and $\beta_{pt}$) and the actual Bowen ratio ($\beta$), i.e., sensible over latent heat, for ocean grid cells/months. (e) The same as (d), but for the differences between the three PETs ($PET_e$, $PET_o$, and $PET_{pt}$) and the ET over the ocean. The mean and uncertainty range of the difference (5th to 95th percentile) are shown for each Bowen ratio and PET as circles and horizontal lines. (f, g) Distribution of the sensitivities of the three Bowen ratios (f) and PETs (g) to the mean temperature, for example $d\beta_w/dT$ and $dPET_e/dT$, which are calculated as the ratio of their differences over the difference in the mean temperature between the wet and driest conditions. The values in parentheses show the mean sensitivities of the three Bowen ratios and PETs for ocean grid cells/months. (For interpretation of the references to color in this figure legend, the reader is referred to the web version of this article.)

---

## Author Comment (AC3)

The authors present an improvement of the so-called complementary relationship (CR). CR is a simplified method to estimate the actual evaporation in a large area under quite restrictive assumptions (outlined in McNaughton and Spriggs, 1989). It is based on the information from an evaporation measurement on a small wet area (saturated surface) within the domain (the authors list pan evaporation explicitly). The aim is to estimate the actual evapotranspiration in the surrounding large area. The authors demonstrate that existing CR variants can be written in a form that contains one empirical parameter, k. The authors' idea is to replace the parameter k with the Bowen ratio of the large area under wet conditions (when potential evapotranspiration applies).

Response:
We appreciate your time and effort in reviewing our manuscript. As noted by McNaughton and Spriggs (1989), although the complementary relationship (CR) has been postulated on physical grounds, it has not yet been derived from first principles. This is primarily due to the ambiguities in the definitions of PETe and PETa, as well as the empirical nature of the parameter k involved.

The main contribution of our study is to develop a physically-based formulation of the CR with explicit definition and estimation of PETe and PETa, and quantitative relationships among ET, PETe, and PETa on the basis of the surface energy balance. Through this process, we have provided an alternative and physically-based interpretation of the empirical parameter k, showing that it corresponds to the wet Bowen ratio ($\beta_w$).

Importantly, our framework does not merely replace the parameter k with $\beta_w$, but derives the complementary relationship from clear definitions of PETe and PETa, leading naturally to a formulation in which $\beta_w$ arises as a physically meaningful parameter rooted in conservation of energy.

The classical CR method is based on a number of simplifying assumptions and controversial. One of the assumptions is that the net radiation is constant. This is not really plausible if the area dries out at the same time. Given that the surface temperature will increase so will the long-wave outward radiation. Yet, since this assumption appears to be common to the CR method, I will take this assumption for granted.

Response:
We acknowledge that the assumption of constant net radiation is fundamental to the complementary relationship (CR), given that its foundation hinges on the surface energy balance, specifically, the redistribution of latent and sensible heat fluxes under contrasting wet and dry conditions. However, when a dry surface becomes saturated, the net radiation may increase or decrease, due to concurrent changes in surface albedo, cloud cover, surface and air temperatures, and other environmental factors (Zhou and Yu, 2024). For example, while a cooler saturated surface may reduce upward longwave radiation, increased cloud shading could simultaneously

diminish downward shortwave radiation. As various factors affect the net radiation in different directions, leading to indeterminate net radiation changes, which cannot be observed unless the whole area becomes saturated in practice. Therefore, the observed net radiation is commonly used to estimate the PET (i.e., PETe in this study).

As you noted, this assumption of constant net radiation remains standard across CR implementations, as it simplifies the quantification of flux partitioning without compromising the framework's core complementary behavior between ET and PETa. Specifically, the proportional coefficient governing this relationship $(-\beta_w)$ remains consistent in both our base formulation (equation (17)), which adopts the constant net radiation assumption, and in a more generalized complementary relationship (equation (A5) in Appendix A) that explicitly accounts for potential changes in net radiation between wet and dry conditions. This consistency confirms that the key complementary dynamics captured by the coefficient $-\beta_w$ are robust in relation to the assumption of constant net radiation, thus supporting the validity of adopting this assumption for our study.

The derivation of the authors is elongated and somewhat meandering. See below for individual comments. To avoid misunderstandings I will briefly summarise the proposed procedure:

1. Calculate the potential evapotranspiration in the large area (PET_e). Calculate the Bowen ratio (beta_w) from the one-dimensional energy balance for the related (wet) conditions.

2. Assume that air temperature and air vapour pressure in the small area are the same as in the large area under real (drying) conditions.

3. Assume that the surface temperature (T_s) in the small area is the same as in the large area under real (drying) conditions. Under the given assumptions (gradient approach, same aerodynamic resistance for both sensible and latent heat fluxes), the sensible heat fluxes (H) in the large and in the small area are equal.

4. Calculate the sensible heat flux in the small area (and thus also in the large area) from given (measured) evapotranspiration (PET_a) in the small area under the assumption that the Bowen ratio in the small area is the same as that in the large area under wet conditions.

5. Given H, calculate the actual evaporation (ET) in the large area from the energy balance.

The assumptions regarding the (a) equal surface temperatures and (b) equality of Bowen ratios regardless of the atmospheric conditions are implausible, if not unphysical. Ad (a): This neglects evaporative cooling. It would mean that an evaporating surface in a dry environment could not be detected with a thermal camera. Ad (b): This assumption contradicts, unnoticed by the authors,

their equation eq. (23). It further contradicts their statement in line 188 that in an experiment the surface temperature would have to be maintained at the appropriate value.

Response:
Thank you for your insightful comments. We appreciate the opportunity to clarify the two assumptions regarding (a) equal surface temperatures and the equality of the wet Bowen ratios, which are critical to the estimation of the potential ET (PETe) and apparent potential ET (PETa) and the derivation of the complementary relationship.

PETe and PETa are hypothetical fluxes that cannot be directly observed and must be estimated based on certain assumptions, as is the standard practice in the literature. While many previous studies estimate PETe from the Priestley-Taylor equation and PETa from the Penman equation, our earlier work (Zhou and Yu, 2024) has demonstrated theoretical and practical problems in these conventional formulations. In this study, we adopted improved formulations introduced in Zhou and Yu (2024) that provide more robust estimates of PETe and PETa and further elaborated their quantitative relationship with ET within the complementary relationship framework (Sections 3.1 and 3.2).

**Section 3.1: Definition and estimation of PETe and PETa**
PETe (also widely known as PET) represents the rate of ET that would occur when water supply is unlimited at the evaporative surface (i.e., a fully saturated surface). As PETe cannot be directly measured unless the whole surface is saturated and must be estimated or derived from observations under dry conditions, the energy-based approach with only two required variables (Rn and $\beta_w$) provide the best approach for PETe estimation with two assumptions.
1) the assumption of constant net radiation (Rn);
2) the assumption of constant wet Bowen ratio ($\beta_w$).

The assumption 1) is commonly adopted in PET estimation, and the assumption 2) has been validated in our previous studies (Zhou and Yu, 2024; 2025), which also demonstrate that PETe is the most reliable PET estimator as $\beta_w$ remains fairly constant due to coupled changes in temperature and humidity of the air and at the land surface from dry to wet conditions. In contrast, the widely used Priestley-Taylor equation for PET estimation shows much larger biases as the implied wet Bowen ratio ($\beta_{pt}$) is sensitive to temperature changes between wet and dry conditions (Zhou and Yu, 2024; 2025). Thus, the second assumption of constant $\beta_{pt}$ for the Priestley-Taylor equation is not strictly valid if it is used to estimate the PET.

PETa represents the ET rate from a small, saturated surface (e.g., a tiny evaporation pan placed in a desert) within a larger, unsaturated area. In this context, the energy available for evaporation comes from both net radiation and heat advection from the surrounding environment. Early studies (e.g., Kahler and Brutsaert, 2006) estimated PETa using pan evaporation measurements. However,

pan evaporation rates are highly sensitive to pan characteristics such as size, material, and exposure, rendering PETa inherently indeterminate and uncertain in practice. This ambiguity poses a major challenge for developing a physically based formulation of the complementary relationship. To address this issue, we estimate the upper limit for PETa by introducing two key assumptions:
i) that the wet area is quite small and has no practical influence on atmospheric conditions.
ii) that the surface temperature of the small wet area is maximized and equals that of the surrounding dry area (Ts).

The assumption i) was explicitly elaborated by Brutsart (2015), which allows the use of observed meteorological variables to estimate PETa, based on the Penman equation. The assumption i) and the Penman equation have been widely used in many CR studies (Szilagyi et al., 2017; Zhang et al., 2017; 2021; Ma et al., 2021). However, the traditional Penman equation neglects the horizontal heat advection from the surrounding environment (Ae), leading to systematic underestimation of PETa (Zhou and Yu, 2024):

$$\lambda PET_a = \frac{\Delta(R_n + A_e) + \frac{\rho c_p (e_a^* - e_a)}{r_a}}{\Delta + \gamma}, A_e > 0$$

Moreover, due to the inherent ambiguity in the definition of PETa and the difficulty in quantifying *Ae*, PETa remains largely indeterminate in practice. To resolve this issue, we adopt the assumption ii) to estimate the upper limit of PETa. This assumption eliminates the ambiguity in PETa and enables direct estimation of PETa using observed and modelled data. In contrast, an uncertain PETa hinders the development of quantitative relationships among ET, PETe, and PETa within the complementary relationship framework. This explains why previous CRs as well as their parameters remain empirical (see the Discussion section 4.1).

We also note that the use of constant surface temperature for estimating PETa has precedent in the literature. Szilagyi (2007) interpreted the Penman equation as representing an extreme case in which the surface temperature of the wet area equals that of the surrounding dry environment—consistent with our assumption ii). Additionally, climate models have adopted a similar approach, estimating PETa (serve as reference conditions for estimating ET) based on prescribed or modeled surface temperature (e.g., Milly, 1992). Our approach provides a consistent method to estimate PETa under observational and model-based conditions, thereby enhancing the applicability of the complementary relationship.

The implementation of Budyko's (1956) scheme commonly found in atmospheric general circulation models has a form that is similar to (4):

$$E = \beta_s E_p(T_s), \qquad (6)$$

but uses a definition of potential evaporation that differs fundamentally from that of Budyko, that is,

$$E_p(T_s) = \frac{\rho}{r_a}[q_s(T_s) - q_a], \qquad (7)$$

in which $T_s$ is the actual computed surface temperature, determined from an energy balance that is similar to (3) but allows only the actual amount of evaporative cooling:

$$R_0 - 4\epsilon\sigma T_a^3(T_s - T_a) - G$$
$$= \beta_s \frac{L\rho}{r_a}[q_s(T_s) - q_a] + \frac{\rho c_p}{r_a}(T_s - T_a). \qquad (8)$$

Here a new subscript is introduced for $\beta$ in (6) since the different definitions of potential evaporation appearing in (4) and (6) can lead to the same rate of evaporation only if the moisture availability functions differ.

[16] The present theory considers two extremes, on one hand, when the heat conduction is perfect and the wet area achieves the same surface temperature as the drying environment, and on the other hand, when there is no energy transfer at all between the two areas having different wetness and therefore different surface temperatures. The application of the Penman equation is a proxy for the perfect heat conduction case as it employs drying environment characteristics. In reality, practical measures of apparent potential evaporation, such as the application of an evaporation pan or an evaporimeter found at standard meteorological stations, may express a degree of asymmetry between (and perhaps beyond for evaporation pans due to differences in surface properties between the pan as well as its water and the surrounding vegetated land) the two theoretical extremes considered.

Note: the left from Szilagyi (2007) and the right from Milly (1992).

**Section 3.2: The relationship among ET, PETe, and PETa**

By clearly defining and estimating PETe and PETa, the complementary relationship naturally emerges from the energy balance equation, which serves as its foundation (equations (16) and (17)). Unlike previous studies that imposed assumed linear or non-linear functional forms among ET, PETe, and PETa (Brutsaert and Parlange, 1998; Szilagyi 2007; Brutsart, 2015; Crago et al., 2016), our derivation requires no such assumptions. Instead, the CR arises directly from conservation of energy and the energy partitioning framework. The two boundaries conditions of the complementary relationship, i.e., ET=PETe=PETa under wet conditions and ET<PETe<PETa when the surface dries up, are satisfied inherently through the definitions and estimation of PETe and PETa. The validity of the derived CR in equation (17) is further supported by observational evidence, as demonstrated using the Fluxnet2015 dataset in Section 3.3.

**Addressing concerns regarding assumptions (a) and (b)**

We fully acknowledge that in reality, evaporative cooling reduces the surface temperature of wet surfaces (assumption a). However, this cooling effect is difficult to quantify consistently because the energy input from the surrounding environment (*Ae*) is often unknown, and the surface temperature of the wet patch becomes indeterminate due to variability in the size and configuration of the wet area. To address this uncertainty and make PETa practically estimable, we adopt the assumption that the surface temperature of the small wet area equals that of the surrounding dry environment. This assumption is not meant to deny the existence of evaporative cooling, but rather to define an upper limit for PETa that can be consistently estimated from observed data without explicitly resolving the issue with *Ae*.

The assumption (b) has been validated by demonstrating that $\beta_w$ calculated with surface temperature in a dry environment remains relatively constant when the whole surface becomes saturated in our previous study (see Figs. 2 and 3 in Zhou and Yu, 2024).

The assumption b) does not contradict with equation (23). The latter provides an empirical estimate of $\beta_w$ as a function of meteorological variables, specifically $\gamma/\Delta$, when direct observations of surface temperature are not available in practice. As noted in the comment on line 55, the CR framework is especially useful when there is limited capacity to explicitly model land-atmosphere feedbacks. By clarifying the meaning of the parameter $\beta_w$, the proposed CR formulation is designed to be more flexible and adaptable to different data availability scenarios. Whether surface temperature (Ts) is available (e.g., from flux towers or reanalysis products) or must be inferred from routine meteorological observations, our framework provides multiple pathways for estimating $\beta_w$, PETe, and PETa.

***Revisions to the Manuscript***
*To address your concerns and prevent potential misunderstandings, we will revise the manuscript as follows:*
*(1) We will revise Section 3.1 to clearly state that our assumptions are based on physical grounds and are invoked to estimate water fluxes, such as PETe and PETa, that could be not measured in practice. Importantly, these assumptions are transparent and physically consistent in comparison to those used in prior approaches—such as the implicit assumptions embedded in the Priestley–Taylor and Penman equations.*
*(2) We will clarify that the assumption of equal surface temperature applies only to the small wet patch under idealized conditions, specifically for estimating the upper limit of PETa. This assumption does not imply a general neglect of evaporative cooling. Rather, it addresses the otherwise indeterminate nature of PETa and enables its estimation based on observed variables.*
*(3) We will emphasize that the validity of constant $\beta_w$ values for estimating PETe has been demonstrated in our previous studies (Zhou and Yu, 2024; 2025). Additionally, we will better explain how approximate estimates of $\beta_w$, when surface temperature observations are unavailable, allow wider application of ET through this new CR framework.*

The authors realised this, see lines 291-296, but they chose to negate the problem. In the Appendix they drop the equality assumption, inserting a correction parameter (rather 1/k2 than k2). They do not discuss that this parameter will of course vary (be state-dependent). They do not realise that even it were constant it would nullify their proposed approach, which is based precisely on the idea of replacing the empirical parameter k with a physically based quantity. The problem of the equal surface temperatures is not resolved either.

I regret this, but in view of the severe physical deficiencies in the manuscript I can only recommend rejecting it.

Response:
Thank you for your feedback. As stated in lines 291–296 of the manuscript, we have clearly summarized the three key assumptions adopted in our study. We respectfully disagree that these assumptions to be problematic and disagree that we "negate the problem".

As explained above, the assumption of equal surface temperatures is introduced to address the otherwise indeterminate nature of PETa. It allows us to estimate an upper limit of PETa from observations, thereby enabling the derivation of the complementary relationship based on the physically meaningful parameter $\beta_w$. This approach does not neglect the occurrence of evaporative cooling over wet surfaces but just provides a practical and consistent reference state necessary for estimating the PETa.

In the Appendix A, we introduced two parameters ($k_1$ and $k_2$) to demonstrate that the derived complementary relationship between ET and PETa is structurally robust, even if Rn and $\beta_w$ are not known or directly measured. These parameters are not intended to be empirically tuned; rather, they are used to show that the complementary relationship between ET and PETa retains its structure—with a proportional coefficient of $-\beta_w$—regardless of possible uncertainties in Rn and $\beta_w$. This reinforces the theoretical integrity of our formulation.

It is also important to emphasize that both expressions, i.e., $(1 + \beta_w)PET_e$ in equation (17) and $k_1(1 + k_2\beta_w)PET_e$ in equation (A5), essentially equal Rn over a large area. Therefore, the consistency of the relationship between ET and PETe is preserved between the two formulations. Thus, for practical applications, equation (17) can be used directly, as it is derived solely from the surface energy balance and employs definitive, physically based estimators for both PETe and PETa. No empirical calibration is needed, and the resulting complementary relationship is easily interpretable.

$$ET = (1 + \beta_w)PET_e - \beta_w PET_a \qquad (17)$$
$$ET = k_1(1 + k_2\beta_w)PET_e - \beta_w PET_a \qquad (A5)$$

Our approach does not "nullify" its purpose of replacing empirical parameters. Traditional complementary relationship relies on parameters like k to fit the data (e.g., Brutsaert and Parlange, 1998) without a clear interpretation of its meaning. Our method does not simply replace the parameter k with $\beta_w$, but rather by first clarifying how PETe and PETa are determined and estimated, and the complementary relationship would follow naturally and emerge from the energy balance equation with a physically meaningful parameter $\beta_w$.

We respectfully disagree that the manuscript has "severe physical deficiencies." Our approach is grounded in surface energy balance, addresses longstanding uncertainties in PETe and PETa estimation and uses $\beta_w$ as a physically meaningful parameter. We are open and transparent with all the assumptions made; the derivation is logical and rigorous, and more importantly, we have shown with strong empirical and observational evidence the clear connection between the wet Bowen ratio $\beta_w$ and parameter k in the CR.

*To address your concerns, we will*
*(1) Revise Section 4.1 to explicitly discuss the rationale for and justification of the three assumptions.*
*(2) Expand the Appendix A to emphasize that the parameter $k_1$ and $k_2$ are not operational parameters, and explicitly state they are illustrative, not used in practical ET estimation, and remove any implication they "correct" the CR.*

We request the opportunity to implement these revisions to strengthen the clarity and rigor of our study.

**Detailed comments**
line 21 will promote

Response:
The phrase "would promote" in the sentence will be revised to "will promote".

55 Are the physical mechanisms really so unclear? I would rather say that we do not have the information, or better, that we run the CR when we do not have the time or resources to model the feedback at the land surface explicitly.

Response:
We sincerely appreciate your insightful perspective, which helps refine our wording. You are correct that the physical mechanisms underlying the CR are not entirely unclear, as prior studies have established core processes such as energy balance constraints and land-atmosphere interactions (e.g., Szilagyi, 2007). Our intended emphasis is that practical limitations (e.g., data scarcity and computational constraints) often hinder the explicit quantification of these mechanisms and identification of the quantitative relationships between the three ET components when applying the CR. For example, the physical meaning of the parameter k involved in the complementary relationship remains unclear, which hinders and limits the application of the complementary relationship for ET estimation. This gap between theoretical understanding and real-world application, we believe, underscores the need for studies like ours to derive a physically-based CR with a meaningful parameter and improve the robustness of CR-based ET estimation.

We will revise the original sentence to reflect this distinction, ensuring it more accurately conveys the practical challenges rather than inherent uncertainty in the mechanisms themselves.

*This is because, while the physical mechanisms underlying the CR are documented, practical challenges—such as ambiguities in the meaning and interpretation of key parameters and difficulties in quantifying relationships between the three types of ET—hinder our ability to derive a physically-based CR for accurate ET estimation.*

69 Why Priestley Taylor? Why not Penman?

Response:
Many previous studies used the Priestley-Taylor equation to estimate PETe and the Penman equation to estimate PETa (e.g., Brutsaert 2015; Zhang et al., 2017; 2021).

72 I would argue that Penman's equation cannot be used if the model domain is not one-dimensional.

Response:
We fully concur with your observation: the Penman equation, designed for one-dimensional (or spatially uniform) systems, does not account for lateral energy exchanges, making it unsuitable for non-one-dimensional domains.

As highlighted in our previous work (Zhou and Yu, 2024), this limitation arises because the original Penman framework only incorporates vertical energy supply via net radiation (Rn), while neglecting horizontal energy transfer from the surrounding environment—a process critical in heterogeneous domains, such as a small patch within a large dry landscape. Consequently, applying the Penman equation would underestimate PETa in such contexts.

To address this, the Penman equation could be modified to include an additional energy term (Ae) to represent the lateral energy inputs:

$$\lambda PET_a = \frac{\Delta(R_n + A_e) + \frac{\rho c_p(e_a^* - e_a)}{r_a}}{\Delta + \gamma}, A_e > 0$$

Notably, *Ae* varies with the size and spatial configuration of the wet surface relative to its surroundings, rendering PETa inherently context-dependent in the domain that is not one-dimensional. This domain dependency implies that PETa (e.g., from a small wet area in a dry environment) cannot be uniquely determined using a universal formulation—one reason why many previous CR formulations remain empirical with empirically defined parameters. To resolve this, our analysis focuses on estimating an upper bound of PETa, defined when the wet surface temperature approaches that of the surrounding environment. This constraint enables the derivation of the CR with a physically meaningful parameter ($\beta_w$).

Introduction

The Introduction would benefit enormously if the authors could clarify in what situation their approach is useful. I had to reconstruct the field of possible application by reading several other papers. This should not happen. In that sense I found that the paper by Szilagyi (2021) is much more instructive.

The target group can be much larger if the introduction is better formulated. For readers unfamiliar with the narrow history of CR, it would help to drop the term "oasis effect".

Response:

We sincerely appreciate your thoughtful feedback, which is instrumental in improving the clarity and accessibility of the Introduction. We fully agree that the manuscript will benefit from a clearer explanation of the practical relevance of our approach and from the use of more broadly accessible language.

In the revised Introduction, we will implement the following improvements:

*1) Clarify practical applications: We will explicitly describe the situations where our approach is most useful, for example, estimating ET in data-scarce regions, enhancing hydrological and land surface models in heterogeneous landscapes, and assessing climate change impacts on regional water availability. These examples will help readers, particularly those unfamiliar with the historical development of the complementary relationship (CR), quickly understand the value of our method.*

*2) Simplify terminology and improve accessibility: In line with your suggestion, we will avoid niche terminology such as "oasis effect," which may not be familiar to a broader audience. We will instead use more intuitive descriptions when introducing the two PET estimators, PETe and PETa, and clearly explain the conceptual distinction between them. We will also emphasize the current gap in defining and estimating both PET components and the need for physically consistent methods to address this issue.*

*3) Broaden the narrative structure: Inspired by the clear and instructive style of Szilagyi et al. (2021), we will restructure the Introduction to begin with broader hydrological and environmental challenges, such as the persistent uncertainties in ET estimation under climate variability and change. before narrowing the focus to our proposed framework. This will better engage a wider audience, including hydrologists, land surface modelers, and water resource managers.*

These revisions aim to make the Introduction more informative, intuitive, and inclusive, ultimately expanding the reach and impact of our study beyond the traditional CR research community.

84 will advance

Response:
The phrase "would advance" in the sentence will be revised to "will advance".

91 Please give units everywhere. Units improve the understanding.

Response:
We will add units for ET, PETe, and PETa (mm/day) in the revised manuscript.

95 Should it not rather read "drying"?

Response:
We will revise "in a dry environment" to "in a drying environment" in the sentence.

98 According to Szilagyi (2007) the symmetry idea comes from Brutsaert and Stricker (1979).

We have carefully checked the literature and can confirm that the concept of the symmetric complementary relationship was originally proposed by Bouchet (1963), and later formally cited and developed by Brutsaert and Stricker (1979). We will cite both references in the revised manuscript.

99 Colon after conditions

Response:
The character "." after conditions will be revised to colon (":").

101 constant instead of "same"

Response:
The phrase "remains the same" in the sentence will be revised to "remains constant".

102, 103 Again: units.

Response:
We will add units for $\lambda$ ($J \cdot kg^{-1}$) and $H_w$ ($J \cdot m^{-2} \cdot s^{-1}$) in the revised manuscript.

103 It would be good to give a reason for the proportionality, in particular with PET_e, not PET_a, in eq. (3). This fundamental assumption should be substantiated in more detail. Or you state that it is just a first approximation (what it probably is).

Response:
The proportionality coefficient reflects the asymmetry for the CR. As noted by Kahler and Brutsaert (2006), this asymmetry arises from the heat transfer between the evaporation pan and its surroundings. Szilagyi (2007) further explored this and showed that the asymmetry originates from the energy dynamics around a small wet surface and not limited only to pan measurements. Specifically, two key processes contribute to this: (a) advection of warmer air over the small wet area; and (b) heat conduction in the ground from the surrounding warmer and dryer soil. These processes enhance the energy supply available for evaporation from the wet area, leading to a nonlinear (asymmetric) relationship between ET and PETa.

In the revised manuscript, we will incorporate this background to clarify that the proportionality in equations (4) and (5) reflects these physical processes. We will also acknowledge that, in the absence of direct energy budget observations, the proportionality is treated as a first-order approximation of the relationship between ET and PETa.

112 Period after "asymmetric"
and better: is widely supported by

Response:
Following your suggestions, the sentence will be revised as follows:

*Equation (4) is identical to the original CR, i.e., equation (3) with $k = 1$, otherwise the CR becomes asymmetric. The latter is widely supported with observations and theoretical derivations (Kahler and Brutsaert, 2006; Szilagyi, 2007, 2021).*

113 is likely to arise
This argument is rather weak, particularly in regard to lines 58,59, where you vaguely state that the symmetric relationship claimed by Bouchet (1963, not accessible) is not supported by observation. Given the complex interplay of processes at the land surface, it is not very probable that a symmetric relationship comes out, depending on the desired accuracy, of course.

Response:
Although Bouchet (1963) is not accessible, its concept of a symmetric complementary relationship has been extensively discussed and cited in subsequent literature, including McNaughton and Spriggs (1989), who summarized and clarified Bouchet's original hypothesis (see the screen capture as insert below).

However, both observational and theoretical studies have since demonstrated that the complementary relationship between ET and PETa is often asymmetric. For example, Kahler and Brutsaert (2006) used observations of actual evapotranspiration and Class A pan evaporation to show that the relationship is asymmetric, primarily due to heat exchange between the pan and its surroundings, as well as local advection effects. Szilágyi (2007) provided further theoretical insight, demonstrating that the degree of asymmetry depends on assumptions about the wet patch's energy balance. In one case, where the wet surface reaches the same temperature as the drying environment due to heat transfer, the relationship becomes highly asymmetric, consistent with our framework for estimating the upper limit of PETa. In contrast, when no external heat transfer is assumed, the relationship becomes more symmetric.

These studies reinforce our argument that a symmetric complementary relationship is not generally supported by observation, and that asymmetry is a more realistic representation of surface energy partitioning under varying conditions. These points will be further clarified and elaborated upon in the revised manuscript.

**THE COMPLEMENTARY RELATIONSHIP**

The arguments presented by Bouchet (1963), and adopted and reworked in a series of papers by Morton (1965, 1969, 1975, 1983), were rather different. Bouchet noted that heat and moisture released at the surface (H and $\lambda E$ respectively) modify the temperature and humidity (T and q respectively) of the air mass above. He argued that the 'potential evaporation' measured over a region is as much the effect of evaporation as its cause. Bouchet's new idea was that the rise in potential evaporation observed above an area as it dries out might actually be used to measure the real evaporation rate from that area.

Bouchet argued: if, for a reason independent of energy supply E is reduced below the potential rate appropriate to a wet region ($E_{po}$) then an amount of energy (Q) would be released so that

$$\lambda E_{po} - \lambda E = Q \tag{3}$$

Since this change within the air mass over the area leaves net radiation almost unaltered, the only important effects will be on temperature, humidity and turbulence, leading to a change in potential evaporation ($E_p$). If the changes do not modify the transfers of energy between the modified air mass and that beyond, this released energy, Q, should just equal the increase in $\lambda E_p$. So without modification of the initial climate from an energy point of view, and especially without modification of the original oasis effects, one has

$$\lambda E_p = \lambda E_{po} + Q \tag{4}$$

whence, with (3)

$$E + E_p = 2 E_{po} \tag{5}$$

Equation (5) is the 'complementary relationship'. It is the central proposition in the CR method. Developments of the idea by Bouchet (1963), Morton (1965, 1983) and Brutsaert & Stricker (1979) have centred on finding suitable definitions for $E_{po}$ and $E_p$.

121 I do not think that the physical meaning of k is unclear. It looks as a lumped parameter that depends on the geometry of small and large areas, on the state of soil-vegetation and atmosphere and the processes therein.

Response:
We respectfully disagree with the statement that the physical meaning of k is clear. In our view, k is an empirical and lumped parameter that is influenced by multiple sources of uncertainty. On one hand, it depends on the geometry and material of the evaporation pan and the surrounding surface wetness conditions, which affect energy exchange processes. On the other hand, it is sensitive to the uncertainties inherent in both PETe and PETa estimation—whether from pan observations or through empirical formulations.

Precisely because k integrates many uncontrolled and context-specific influences, its physical interpretation remains ambiguous. Determination or estimation of k requires fitting data to the assumed CR. Our analysis demonstrates that, if PETe and PETa are defined and estimated using equations (12) and (15), k can be interpreted as the wet Bowen ratio ($\beta_w$). This physically meaningful interpretation, to our knowledge, has not been explicitly stated in previous literature.

eqs (6,7), 137 You should not simply an established symbol like R_n and call it the difference between its actual meaning and the ground heat flux. The minimum is to call it R_n^* or R`_n or similar.
You should not call the difference between net radiation and ground heat flux net radiation. It is often called available energy.

Response:
Thank you for the helpful clarification. We agree with your suggestion and will replace $R_n$ with $R'_n$ to explicitly denote the available energy, defined as the difference between net radiation and ground heat flux, to avoid potential confusion. In the original manuscript, we referred to this quantity as "net radiation" for simplicity, since ground heat flux is typically negligible at monthly and longer time scales. While the term "available energy" is a broad term and can sometimes include horizontal energy transfers (especially relevant for PETa), we will clarify its usage in the revised manuscript as follows:

*where $R'_n$ ($J \cdot m^{-2} \cdot s^{-1}$) denotes the available energy, i.e., net radiation minus ground heat flux (hereafter referred to as net radiation for simplicity, as ground heat flux is negligible at monthly or longer time scales), which equals the sum of latent and sensible heat fluxes.*

133, 134 heat flux. Again, give the units, this will make the text more readable.

Response:
We will add units for these energy fluxes ($J \cdot m^{-2} \cdot s^{-1}$) in the revised manuscript.

138 Even this is a simplification already.
flux

Response:
Thank you for pointing this out. In the revised manuscript, we will clarify that the term "net radiation" is a simplification and explicitly refer to the fluxes involved. For details, please refer to our response and revision to the comment on line 137.

Chapter 3.1
You invoke the impression that these so-called approaches are somehow equivalent, but they are not at all. Eqs (6,7) are equivalent and state the conservation of energy at the land surface (simplified, but this is ok in this context). The equations are redundant and cannot be solved for beta. So this is not an "approach".

Eqs (8,9) are process equations. In combination, they give a description of what is happening at the land surface and, given the parameter values, can be solved for the turbulent fluxes.
I am sure that you know all this, it is just the presentation that makes the derivation crude.
The meaning of the indices is not explained.

Response:
We agree that the energy balance and aerodynamic formulations are not equivalent in a strict methodological sense. Our intention was not to present them as such, but rather to highlight that they should yield consistent estimates of the latent and sensible heat fluxes.

Equations (6) and (7) represent the expressions of latent and sensible heat fluxes based on the surface energy balance. These equations serve as a conceptual foundation for understanding energy partitioning at the land surface (Fig. 1). The energy balance approach can be used to estimate latent and sensible heat fluxes when the Bowen ratio is observed, such as in the Bowen Ratio Energy Balance System (see EBBR Handbook). Additionally, the Bowen ratio can be inferred from the aerodynamic formulation (as shown in equation (10)), or estimated empirically, particularly under

wet conditions (for example, the Priestley-Taylor equation). Therefore, we believe it is appropriate to refer to this as the energy balance approach.

Given that both the conceptual background and practical application of these two approaches are well established in the literature (e.g., Penman, 1948; Chow et al., 1988; Bonan 2008; Zhou and Yu, 2024), we did not include detailed derivations in the manuscript. However, we will cite the relevant references and clarify the distinctions more explicitly in the revised version.

150 This is a distorted formulation for combining process equations with the mass balance equation.

Response:
We respectfully disagree with this comment. Estimating the Bowen ratio from the aerodynamic formulation is a widely used and accepted approach for enabling the application of the energy balance method (i.e., equations (6) and (7)). This formulation has been adopted in many foundational studies and textbooks (e.g., Penman, 1948; Chow et al., 1988; Zhou and Yu, 2024), and provides a practical means to integrate process-based understanding with energy and mass balance constraints.

141 It is the flux, everywhere.
You should give a textbook reference here for interested readers.

Response:
Thank you for the suggestion. We will clarify in the revised manuscript that the terms refer to latent and sensible heat fluxes throughout the manuscript. Additionally, we will cite Bonan (2008) as a textbook reference for details.

160 This would require knowledge of T_s. This should at least be mentioned.
partitioning

Response:
We agree that the estimation of $\beta_w$ requires knowledge of surface temperature ($T_s$). This dependency has been noted in line 157, and we will ensure that it is clearly stated and emphasized in the revised version.

163 What is eq. (13) needed for?

Response:
Equations (12) and (13) describe the partitioning of latent and sensible heat fluxes under wet conditions, in contrast to the dry-condition partitioning represented by equations (6) and (7).

Equation (13) is specifically referenced in line 222 to explain the assumption of constant net radiation when comparing wet and dry conditions. These distinct partitioned regimes are also illustrated in Fig. 1 to clarify their roles within our framework, as the complementary relationship arises from shifts in the partitioning between latent and sensible heat fluxes between wet and dry conditions.

170 are assumed to be identical

Response:
The sentence will be revised accordingly.

188 In lines 171,172 you assumed that the surface temperatures are equal. It is a contradiction that you have to maintain the temperature in an experiment.

Response:
We assumed that the surface temperature of the small, saturated area approaches the temperature of the surrounding dry surface ($T_s$), sustained by heat transfer from the environment. This assumption allows us to estimate the upper limit of PETa using equation (14) or (15).

In practice, the Class A evaporation pan (121 cm in diameter) has a relatively large surface area, and its surface temperature is often lower than that of the surrounding environment due to evaporative cooling. Therefore, to approximate the upper limit of PETa using a pan, we need to maintain its surface temperature equal to that of the surrounding dry surface. This is consistent with the assumption used in our framework for estimating the upper limit of PETa, rather than contradicting it.

254 Why did you not use the measured Rn? To avoid dealing with the energy balance closure problem? Then express this clearly. Were the turbulent flux data not Bowen-ratio corrected?

Response:
In our analysis, we defined Rn as the sum of latent and sensible heat fluxes, consistent with equations (6) and (7). For simplicity and consistency, we directly used the sum of latent and sensible heat fluxes to calculate Rn, and subsequently PETe and PETa, at fluxnet sites.

We did not use the measured Rn directly, nor did we apply the Bowen ratio correction method, which redistributes latent and sensible heat fluxes in proportion to the observed Bowen ratio. Our focus is not on closing the energy balance *per se*, but rather on examining the relationships between scaled ET (ET/PETe) and scaled PETa (PETa/PETe), and between scaled ET (ET/PETa) and scaled PETe (PETe/PETa) (Fig. 2). As summarized in Table 2, these relationships are fundamentally governed by the Bowen ratio ($\beta$) and the wet Bowen ratio ($\beta_w$). Therefore, whether

or not energy balance closure is enforced via the Bowen ratio correction method does not affect the validity of the relationships among ET, PETe, and PETa in our study so long as the Bowen ratio remains consistent and unchanged.

255 To do this (eq. 11), you would need the surface temperature. More details are necessary here.

Response:
We clarified in the manuscript that surface soil temperature was used as a proxy for surface temperature in our analysis. This will be explicitly stated in the revised manuscript as follows:

*Data were included in this analysis for site-years where measured or high-quality gap-filled data on air temperature, surface soil temperature (used as a proxy for surface temperature), sensible and latent heat fluxes were available.*

415 Given the validity of eq. 23, it shows, apparently unnoticed by the authors, that beta_w is a function of Delta and hence of air temperature.

Response:
Indeed, as shown in equation (23), $\beta_w$ is approximately estimated as a function of $\gamma/\Delta$, which implicitly links it to air temperature. Equation (11) defines $\beta_w$ based on aerodynamic principles and requires surface temperature data, which may be unavailable in many practical applications. To address this limitation and enhance the applicability of our framework, we introduced equation (23) as an empirical approximation of $\beta_w$, based on the relationship between $\beta_w$ and $\gamma/\Delta$ derived using ocean data (Yang and Roderick, 2019; Zhou and Yu, 2024). This formulation provides a practical means to estimate $\beta_w$ using standard meteorological variables and facilitates broader application of the complementary relationship framework, particularly in settings where surface temperature is not observed.

**References:**
Bonan, G. B. (2008). Ecological Climatology: Concepts and Applications (2nd ed.). Cambridge University Press.
Bouchet, R. J.: Evapotranspiration reelle, evapotranspiration potentielle, et production agricole, Annales Agronomiques, 14, 743–824, 1963.
Brutsaert, W.: A generalized complementary principle with physical constraints for land-surface evaporation, Water Resour. Res., 51, 8087–8093, 2015.
Brutsaert, W. and Parlange, M. B.: Hydrologic cycle explains the evaporation paradox, Nature, 396, 30–30, https://doi.org/10.1038/23845, 1998.
Brutsaert, W. and Stricker, H.: An advection-aridity approach to estimate actual regional evapotranspiration, Water Resour. Res., 15, 443–450, 1979.
Chow, V. T., Maidment, D. R., and Mays, L. W.: Applied Hydrology, International Edition., McGraw-Hill Book Company, New York, 1988.

Crago, R., Szilagyi, J., Qualls, R., and Huntington, J.: Rescaling the complementary relationship for land surface evaporation, Water Resources Research, 52, 8461–8471, 2016.

Kahler, D. M. and Brutsaert, W.: Complementary relationship between daily evaporation in the environment and pan evaporation: DAILY AND PAN EVAPORATION, Water Resour. Res., 42, 2006.

Ma, N., Szilagyi, J., and Zhang, Y.: Calibration-Free Complementary Relationship Estimates Terrestrial Evapotranspiration Globally, Water Resources Research, 57, 2021.

McNaughton, K. G., and Spriggs, T. W.: An evaluation of the Priestley and Taylor equation and the complementary relationship using results from a mixed-layer model of the convective boundary layer, Estimation of areal evapotranspiration, 177, 89-104, 1989.

Milly, P. C. D.: Potential evaporation and soil moisture in general circulation models, Journal of climate, 5(3), 209-226, 1992.

Penman, H. L.: Natural evaporation from open water, bare soil and grass, Proceedings of the Royal Society of London. Series A: Mathematical and Physical Sciences, 192, 120–145, 1948.

Priestley, C. H. B. and Taylor, R. J.: On the Assessment of Surface Heat Flux and Evaporation Using Large-Scale Parameters, Mon. Wea. Rev., 100, 81–92, 1972.

Szilagyi, J.: On the inherent asymmetric nature of the complementary relationship of evaporation, Geophys. Res. Lett., 34, L02405, 2007.

Szilagyi, J.: On the thermodynamic foundations of the complementary relationship of evaporation, Journal of Hydrology, 593, 125916, 2021.

Szilagyi, J., Crago, R., and Qualls, R.: A calibration-free formulation of the complementary relationship of evaporation for continental-scale hydrology, Journal of Geophysical Research: Atmospheres, 122, 264–278, 2017.

Yang, Y. and Roderick, M. L.: Radiation, surface temperature and evaporation over wet surfaces, Q.J.R. Meteorol. Soc., 145, 1118–1129, 2019.

Zhang, L., Cheng, L., and Brutsaert, W.: Estimation of land surface evaporation using a generalized nonlinear complementary relationship, J. Geophys. Res. Atmos., 122, 1475–1487, 2017.

Zhang, L. and Brutsaert, W.: Blending the Evaporation Precipitation Ratio With the Complementary Principle Function for the Prediction of Evaporation, Water Resources Research, 57, 2021.

Zhou, S. and Yu, B.: Physical basis of the potential evapotranspiration and its estimation over land, Journal of Hydrology, 641, 131825, 2024.

Zhou, S. and Yu, B.: Reconciling the Discrepancy in Projected Global Dryland Expansion in a Warming World, Global Change Biology, 31, e70102, 2025.